



# EURADCLIM: The European climatological high-resolution gauge-adjusted radar precipitation dataset

Aart Overeem[1], Else van den Besselaar[1], Gerard van der Schrier[1], Jan Fokke Meirink[1], Emiel van der Plas[1], and Hidde Leijnse[1]

[1]Royal Netherlands Meteorological Institute, Utrechtseweg 297, 3731 GA De Bilt, The Netherlands

**Correspondence:** Aart Overeem (overeem@knmi.nl)

**Abstract.** The European climatological high-resolution gauge-adjusted radar precipitation dataset, EURADCLIM, addresses the need for an accurate (sub-)daily precipitation product covering 78% of Europe at high spatial resolution. A climatological dataset of 1-h and 24-h precipitation accumulations on a 2-km grid is derived for the period 2013 through 2020. The starting point is the European Meteorological Network (EUMETNET) Operational Program on the Exchange of weather RAdar Information (OPERA) gridded radar dataset of 15-min instantaneous surface rain rates, which is based on data from, on average, 138 ground-based weather radars. First, methods are applied to further remove non-meteorological echoes from these composites by applying two statistical methods and a satellite-based cloud type mask. Second, the radar composites are merged with the European Climate Assessment & Dataset (ECA&D) with potentially ~7700 rain gauges from National Meteorological and Hydrological Services (NMHS) in order to substantially improve its quality. Characteristics of the radar, rain gauge and satellite datasets are presented, as well as a detailed account of the applied algorithms. The clutter removal algorithms are effective, while removing few precipitation echoes. The usefulness of EURADCLIM for quantitative precipitation estimation (QPE) is confirmed by comparing against rain gauge accumulations employing scatter density plots, statistical metrics, and a spatial verification. These show a strong improvement with respect to the original OPERA product. The potential of EURADCLIM to derive pan-European precipitation climatologies and to evaluate extreme precipitation events is shown. Specific attention is given to remaining artefacts in and limitations of EURADCLIM. Finally, it is recommended to further improve EURADCLIM by applying algorithms to 3D instead of 2D radar data, and by obtaining more rain gauge data for the radar-gauge merging. The EURADCLIM 1-h and 24-h precipitation datasets are publicly available at https://doi.org/10.21944/7ypj-wn68 (Overeem et al., 2022a) and https://doi.org/10.21944/1a54-gg96 (Overeem et al., 2022b).

## 1  Introduction

Accurate surface precipitation information at high spatiotemporal resolutions is needed for many scientific domains and applications, such as agriculture, water management, weather prediction and climate monitoring, but is often lacking. EUMETNET (European Meteorological Network) is a network of 31 National Meteorological and Hydrological Services (NMHS) and one of its programs is OPERA (Operational Program on the Exchange of weather RAdar information). In OPERA, expertise on operational ground-based weather radars is exchanged and pan-European radar products have been developed, which are dis-



seminated in near real-time (Huuskonen et al., 2014; OPERA, 2022). While the EUMETNET OPERA ground-based weather radar composite provides strong coverage at the km scale, it generally underestimates precipitation by tens of percent. The spatial variability of this bias indicates that its quality is inhomogeneous in time and space. Moreover, many smaller areas suffer from severe overestimation due to non-meteorological echoes (clutter), mainly due to signal interference (Saltikoff et al., 2016), obstacles in the vicinity of radars, and refraction of the radar beam (e.g. Gourley et al., 2007; Fabry, 2015; Overeem et al., 2020). A long list of possible sources of error can negatively affect radar precipitation estimates (Doviak and Zrnić, 1993; Fabry, 2015; Rauber and Nesbitt, 2018; Ryzhkov and Zrnic, 2019): for instance, hardware-related errors such as electronic calibration and antenna pointing offsets (Frech et al., 2017), and severe underestimation due to rain-induced attenuation along the radar beam for X- or C-band radars (e.g. Hitschfeld and Bordan, 1954; Tabary et al., 2009; Jacobi and Heistermann, 2016; Overeem et al., 2021). Another source of error are, for instance, changes in the vertical profile of reflectivity, where the height of the radar sampling volume increases with increasing range from the radar, hence becoming less representative for the reflectivity at the ground (Hazenberg et al., 2013). In contrast, rain gauges often provide accurate local quantitative precipitation estimation (QPE), but their network densities are usually too sparse to capture the spatial precipitation variability, especially at the sub-daily time scale (Van de Beek et al., 2012). Gridded precipitation datasets based on interpolated gauge accumulations and covering large parts of Europe, provide at best daily accumulations at 0.1° and 0.25° grids (Cornes et al., 2018).

A common practice on a national level is to combine the best of two worlds by merging radar with rain gauge data (e.g. Holleman, 2007; Goudenhoofdt and Delobbe, 2016; Nelson et al., 2016; Fabry et al., 2017; Bližňák et al., 2018; Winterrath et al., 2018; Barton et al., 2020). For Europe, Park et al. (2019) developed an operational OPERA-based radar rainfall product for the European Rainfall-InduCed Hazard Assessment (ERICHA) system, employing data from ∼1500 rain gauges. This is used to compute flash flood hazard for Europe for the next 5 hours for the European Flood Awareness System (EFAS). The use of this dataset is restricted to EFAS.

Here, we present an open climatological OPERA-based radar precipitation product over the period 2013–2020, called EU-RADCLIM (EUropean RADar CLIMatology). It covers ∼$8 \times 10^6$ km$^2$ of Europe, which is about 78% of geographical Europe, and covers a variety of climates from Mediterranean to temperate, mountain, continental and arctic. Some differences with the study by Park et al. (2019) are that additional algorithms to remove non-meteorological echoes are applied, and data from far more rain gauges are available ( ∼7700). In addition, for each 1-hour interval that is adjusted, the corresponding gauge data are used to compute a spatial adjustment factor field for that hour, instead of applying such a field based on radar and gauge data from the last 7 rainy days, as is the case in the Park et al. (2019) dataset. In EURADCLIM, non-meteorological echoes are further removed by applying an open-source statistical filter taking into account large gradients and the size of contiguous echoes (Gabella and Notarpietro, 2002; Wradlib, 2021). Since it could also be applied in (near) real-time, its evaluation is also relevant for the existing gridded OPERA products. Next, a climatological satellite cloud type product is employed to identify areas with semitransparent clouds or without clouds, and set rain rates to 0 mm h$^{-1}$ in those areas. Finally, for each year a static clutter mask is computed based on outliers in annual precipitation. For these locations, 1-h radar precipitation accumulations are replaced by spatially interpolated values. Merging these radar precipitation accumulations with those from rain gauges, re-





sults in EURADCLIM, which can be seen as the continental European analogue of a climatological radar precipitation dataset developed for the United States (Fabry et al., 2017).

The outline of this paper is as follows: First, characteristics of the employed radar, rain gauge and satellite datasets, such as data availability and coverage, are described (Sect. 2). Next, the three algorithms to remove non-meteorological echoes and the radar-gauge merging algorithm are presented (Sect. 3). This is followed by a step-by-step evaluation of the datasets after

each processing step, including the EURADCLIM dataset, against rain gauge data (Sect. 4.1–4.2). EURADCLIM's limitations are discussed and illustrated together with pan-European precipitation climatologies (Sect. 4.3). The results section ends with extreme precipitation events derived from EURADCLIM (Sect. 4.4). Finally, conclusions and recommendations to improve EURADCLIM are provided (Sect. 5).

## 2   Data

### 2.1   OPERA radar data

The radar composite product "instantaneous surface rain rate" was obtained from the EUMETNET OPERA Data Centre (ODC or Odyssey) from the period 2013–2020. It is stored in Hierarchical Data Format version 5 (HDF5) files employing the OPERA Data Information Model (ODIM) (Michelson et al., 2019). This product has a temporal resolution of 15 min and a spatial resolution of 2 km $\times$ 2 km (Lambert Azimuthal Equal Area projection; $2200 \times 1900$ grid cells). As is usually the

case for gridded radar precipitation products, the effective resolution decreases for increasing distances from radars and will become lower than 4 km$^2$ (typically at $\sim 115$ km from a radar for a beam width of $1°$). It is based on the raw single site radar data, which have undergone Doppler clutter filtering. Depending on the radar, beam-blockage correction and additional (dual-polarization) clutter filtering have been applied by the respective NMHS. OPERA applies algorithms on either individual radar data or the composite, concerning further removal of non-meteorological echoes and, since late 2015, beam-blockage

correction and a satellite cloud mask to remove non-meteorological echoes (Saltikoff et al., 2019b).

The number of contributing radars, on average 138 based on intervals with data, gradually increases over the period 2013–2020 (Fig. 1a). A variety of radars is employed, for instance, different manufacturers, different frequencies (mostly C-band, some X-band and S-band) and a mixture of single-polarization and dual-polarization radars. The measurement frequency of the radars is 5 min or 10 min, and data from the last time stamp are used in the composite product, e.g., the 5-min file from

10:15 UTC for the 10:15 UTC OPERA composite product. The lowest elevation scan data from all radars are combined to produce a composite of gridded horizontally polarized radar reflectivity factor ($Z_{\mathrm{h}}$) data. Before 29 Sep 2017 08:52 UTC, this was done via logarithmic range-weighted averaging (dB$Z_{\mathrm{h}}$) and afterwards via linear range-weighted averaging ($Z_{\mathrm{h}}$). Finally, instantaneous surface rain rates are retrieved from the reflectivity composites every 15 min using the Marshall-Palmer $Z_{\mathrm{h}} - R$ relation ($Z_{\mathrm{h}} = 200R^{1.6}$). Saltikoff et al. (2019b) provide more details on the OPERA radar data and its processing.

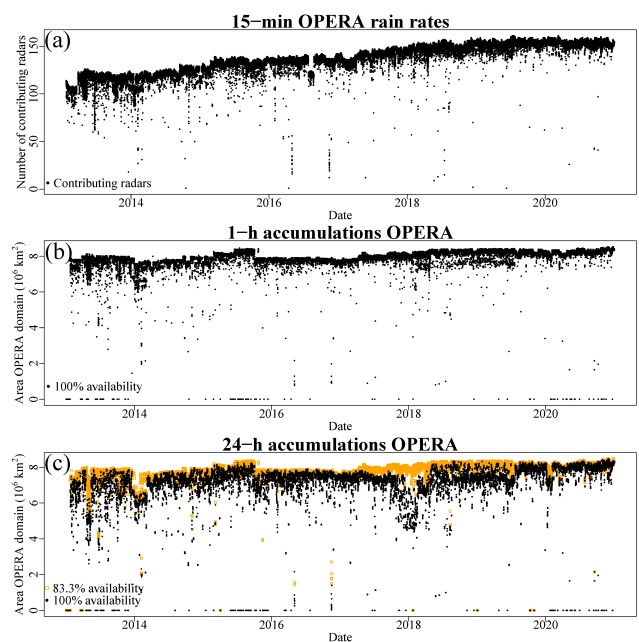

**Figure 1.** OPERA radar data availability as a function of time over the period 2013–2020. (a) Number of contributing radars to the OPERA composite of rain rates in case data are available for at least part of the OPERA domain. Coverage of OPERA radar datasets as a function of time for the (b) 1-h and (c) 24-h precipitation accumulations. For (b) and (c) also time intervals without data are plotted (as 0).

For each radar grid cell (pixel) and clock-hour (i.e., every hour on the hour), 1-h precipitation accumulations are computed from the rain rates in case of full availability, otherwise the grid cells contain the ODIM "nodata" value, which is "used to denote areas void of data (never radiated)" (Michelson et al., 2019). And "undetect" values, "used to denote areas below the measurement detection threshold (radiated but nothing detected)" (Michelson et al., 2019), are set to 0 mm. These 1-h precipitation accumulations are used to compute 24-h accumulations every clock-hour as well as annual accumulations. For each radar grid cell, a minimum data availability of underlying 1-h accumulations of 83.3% is demanded (i.e., at least 20 of 24 hours, or ∼304.2 of 365 days). Grid cells with too low availability are set to the OPERA "nodata" value. The availability of 1-h and 24-h precipitation accumulations is generally high (Fig. 1b–c). For the large majority of the OPERA domain, availability of 1-h precipitation accumulations is at least 95% over the period 2013–2020 (Fig. 2a). The distance to the nearest radar displays quite some variability, but is generally shorter than 175 km (Fig. 2c). The median and mean distance to the nearest radar is 110 km and 133 km, respectively. Some countries do have radars but these do not contribute to the OPERA composite yet (e.g., Austria and Italy). All derived radar datasets are kept in ODIM-HDF5 format on the default OPERA grid of 2 km resolution.

## 2.2 ECA&D and E-OBS rain gauge data

Daily precipitation series were obtained mid June 2022 from the European Climate Assessment and Dataset (ECA&D, https://www.ecad.eu) project (Klein Tank and coauthors, 2002; Klok and Klein Tank, 2008). In total, ∼7700 rain gauges from 29





**Figure 2.** (a) Map of the OPERA domain with fraction of available radar composites over the period 2013–2020. (b) Map with combined radar-gauge availability over the period 2013–2020. (c) Map of distance to nearest radar per grid cell assuming full availability of radar data (note that some of the radars only contributed part of the period). (d) Map of distance to nearest rain gauge per grid cell assuming full availability of radar and gauge data. This shows the best possible result. In reality, the average minimum distance will be longer because sometimes gauge data are missing. Maps made with Natural Earth. Free vector and raster map data ©naturalearthdata.com.





different countries and 37 different data providers, including non-downloadable series (i.e., included in ECA&D for production of derived data but only accessible through the data-owning NMHS), are covered by OPERA radars during (part of) the 2013–2020 period. Combined radar-gauge data availability is at least 90% for most regions (Fig. 2b). Figure 2d displays the distance to the nearest gauge for the OPERA domain if all ∼7700 gauges would be available. The large variability in space of the underlying rain gauge network density is obvious. The median and mean distance to the nearest rain gauge is 42 km and 92

km, respectively. These relatively long distances are mainly caused by areas above land surface with low rain gauge network densities, and above sea. The ECA&D rain gauge dataset will be used to evaluate the various radar precipitation datasets and will be merged with 1-h radar accumulations.

The rain gauge data have undergone quality control by the ECA&D team (Project team ECA&D, Royal Netherlands Meteorological Institute KNMI, 2021) and often by NMHS. Given the latency in gauge data provided to ECA&D for some networks

and to prevent spatial differences in the quality of merged radar-gauge QPE, only data up to and including 2020 are used. At the time of data production (mid June 2022) it was found for some countries that the density of gauge networks from which data were available is relatively sparse (Bosnia and Herzegovina, Croatia, Denmark, Hungary, Iceland, Lithuania, Portugal, Romania, Slovakia, Spain, Switzerland), no data are available (e.g., Bulgaria, Greece, Kosovo, North Macedonia) or not complete for the entire period (e.g., for Romania it ends 31 September 2020, for Serbia it ends 31 December 2017, only a few years

for Montenegro, and time series from most stations in the United Kingdom end 31 December 2019).

The daily measurement interval of gauges is often not exactly known to ECA&D. For instance, the metadata for some networks is imprecise as aggregation intervals ending at 06:00, 07:00 or 08:00 UTC are lumped together. Sometimes, additional information on the measurement interval end time from the respective NMHS was available and selected. In order to determine the exact measurement interval for other gauge networks, gauge accumulations are compared to OPERA 24-h accumulations

by testing different measurement interval end times. For each network, distributions of correlation coefficients for all gauge locations are evaluated for each interval end time using Ridgeline plots. The measurement interval end time with the highest correlations is selected for that given network. The end times of the observations display a large variability across Europe from 0 UTC (9 networks), 6 UTC (16 networks), 6 UTC in summer and 7 UTC in winter (1 network), 7 UTC (1 network), 8 UTC (3 networks), 9 UTC (2 networks), 18 UTC (3 networks), 21 UTC (1 network), to 22 UTC in summer and 23 UTC in winter

(1 network).

A pan-European dataset, E-OBS version 25.0e (https://surfobs.climate.copernicus.eu/dataaccess/access_eobs.php#datafiles), of gridded, daily, interpolated ECA&D gauge accumulations (Cornes et al., 2018) is used to compute annual precipitation accumulations. These will be used for comparison with EURADCLIM accumulations.

## 2.3  Satellite cloud type product

Information on the occurrence and type of clouds was obtained from the Spinning Enhanced Visible and Infrared Imager (SEVIRI) onboard the geostationary Meteosat Second Generation (MSG) satellites operated by the European Organisation for the Exploitation of Meteorological Satellites (EUMETSAT). The CLoud property dAtAset using SEVIRI edition 2 (CLAAS-2) was used, produced by EUMETSAT's Satellite Application Facility on Climate Monitoring (CM SAF). CLAAS-2 (Finken-





sieper et al., 2016; Benas et al., 2017) is a climate data record of cloud properties derived from SEVIRI measurements and
extending from 2004 to present. The CLAAS-2 cloud type product was derived with the MSGv2012 software package developed by the SAF on Nowcasting and Very Short Range Forecasting (NWC SAF). Further details on the retrieval algorithm
can be found in Derrien and Le Gléau (2005) and NWC SAF (2013). The temporal resolution of CLAAS-2 is 15 minutes. Its
spatial resolution is 3 km × 3 km at the sub-satellite point and around 4 km × 7 km in the centre of the OPERA domain ($\sim 52°$
N). The cloud type was converted from the CLAAS-2/SEVIRI grid to the native OPERA radar grid of 2 km by 2 km using
nearest neighbour resampling. This dataset is available day and night and is used to remove non-meteorological echoes from
the OPERA radar data. Over the period 2013-2020, 99.7% of the 15-min intervals have valid data.

## 3  Methodology

The flowchart in Fig. 3 provides an overview of the input datasets, the applied processing, and the output dataset EURADCLIM.
Three algorithms are applied to further remove non-meteorological echoes from the OPERA radar data. Finally, the radar data
are merged with the ECA&D rain gauge data.

### 3.1  Gabella clutter filter

The function *clutter.gabella* from the open-source Python library for weather radar data processing wradlib version 1.9.0
(Heistermann et al., 2013; Mühlbauer et al., 2020) is employed to classify non-meteorological echoes. This Gabella filter is a
two-part algorithm (Gabella and Notarpietro, 2002; Wradlib, 2021), which uses as input the radar reflectivity factors. For this,
rain rates are converted to radar reflectivity factors employing $Z_{\mathrm{h}} = 200R^{1.6}$ (beforehand "undetect" and "nodata" values are
set to 0 mm h$^{-1}$). Then the Gabella filter is used to classify grid cells using the Cartesian radar reflectivity factor data. In the
first part of the filter, strong spatial gradients are identified by checking for each grid cell how many cells surrounding it in a
square lattice of $5 \times 5$ cells are less than 6 dB$Z_{\mathrm{h}}$ lower than the central cell. When this number of cells is lower than 6, the
central cell is identified as clutter. In the second part of the filter, the ratio between the area and circumference for contiguous
echo regions is computed, where these consist of cells with a value above 0 dB$Z_{\mathrm{h}}$. When the absolute value of this ratio is
lower than 1.3, the central cell is identified as clutter. Next, the original surface rain rates are set to 0 mm h$^{-1}$ in case one or two
of the parts of the Gabella filter identify clutter. Central cells that have "nodata" values in the original rain rates are unaffected
by the Gabella filter, and "undetect" values are kept in case of no clutter. A successful example of applying the Gabella clutter
filter for the Netherlands and surroundings is provided in Fig. 4a–b.

### 3.2  Satellite cloud type mask clutter filter

A satellite cloud type mask is employed to classify remaining non-meteorological echoes. Localisation errors (e.g., advection,
timing differences between radar and satellite, parallax) are taken into account by considering all grid cells in a square lattice
of $7 \times 7$ cells containing the central cell for which the classification is performed. The rain rate for this central cell is set
to 0 mm h$^{-1}$ when all cells in the square lattice are either classified as cloud-free or as containing semitransparent clouds





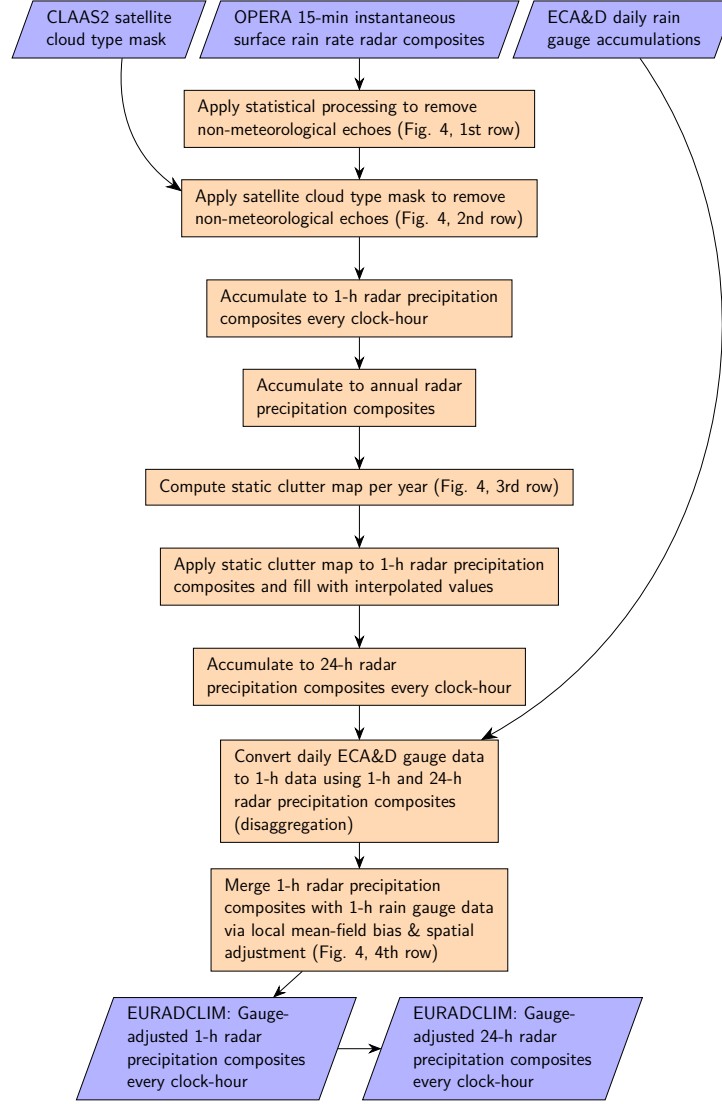

**Figure 3.** Flowchart of radar and gauge data processing for EURADCLIM.

(which are assumed to be non-raining). Concretely, these cases correspond to the following MSGv2012 cloud type categories: cloud free land, cloud free sea, land contaminated by snow, sea contaminated by snow/ice, high semitransparent thin clouds, high semitransparent meanly thick clouds, and high semitransparent thick clouds. In case satellite data is not available for a given pixel (category "non-processed containing no data or corrupted data" or no file/image available), that pixel is not used as a neighbouring pixel and the radar pixel directly beneath it will not be labelled as clutter. Figure 4c–e illustrates the

working of this algorithm by providing a 15-min radar rain rate, before and after applying the satellite cloud type mask, which is also displayed. Because the satellite images are referenced to with the start time of observation and the radar composites





are referenced to with the end time of observation, a satellite image of, for example, 12:00 UTC is combined with the radar composite of 12:15 UTC.

### 3.3 Static clutter filter

The 15-min radar rain rates are accumulated to 1-h and these to annual precipitation accumulations. The function *clutter.histo_cut* from the open-source Python library for weather radar data processing wradlib version 1.9.0 (Heistermann et al., 2013; Mühlbauer et al., 2020; Wradlib, 2022) is employed to classify non-meteorological echoes in the annual precipitation accumulation. First, a histogram of 50 classes is computed from the annual precipitation from a given year using all grid cells. Next, the class with the largest frequency is determined. An iterative procedure detects those classes with a frequency below

5% of the frequency of the class with the largest frequency (the hard-coded default value is 1%). The procedure stops when the changes in the maximum annual rainfall from the remaining classes becomes (smaller than) 1 mm compared to the previous iteration. The grid cells corresponding to the detected classes become the static clutter mask. For each year a separate static clutter mask is obtained. This algorithm identifies areas with static clutter and may also detect areas affected by beam-blockage. Next, inverse distance weighted interpolated values are computed for the identified grid cells (inverse distance weighting power

of 2; maximum of 4 neighbours). This value replaces the original value only in case it is lower than the original value. The original "nodata" values are kept in the output dataset. The interpolation is performed on the 1-h precipitation accumulations. From the cleaned 1-h accumulations, which are used for merging with rain gauge accumulations, 24-h accumulations for every clock-hour are derived. Figure 4f–h shows an example of an annual precipitation accumulation, the derived annual precipitation accumulations after applying the static clutter mask including interpolation on the underlying 1-h radar precipitation compos-

ites, and the corresponding static clutter mask. The mask seems to correspond mostly to areas with high annual precipitation. In theory, the interpolation could decrease precipitation estimates for areas with beam-blockage. Areas with beam-blockage are not abundant, though. Also note that a beam-blockage correction has already been applied by OPERA.

**Figure 4.** Illustration of application of all three clutter removal steps and the rain gauge adjustment, i.e., of all processing steps in EU-RADCLIM. (a)–(b) Application of the Gabella clutter filter (map data ©OpenStreetMap contributors 2022. Distributed under the Open Data Commons Open Database License (ODbL) v1.0), and of (c)–(e) the cloud type mask to remove non-meteorological echoes from a 15-min OPERA composite of rain rates. For the cloud type mask, the grey areas indicate where the cloud type does not belong to the seven categories listed in Sect. 3.2, and for which the radar rain rates were thus left untouched. (f)–(h) Illustration of application of the static clutter mask, derived from the annual precipitation map, to 1-h radar precipitation composites, which are accumulated to an annual precipitation map. (i)–(k) Application of the radar-gauge merging algorithm going from unadjusted to adjusted 1-h rainfall accumulations employing the 1-h adjustment factor field. (c)–(k) Maps made with Natural Earth. Free vector and raster map data ©naturalearthdata.com.





## 3.4 Merging radar with ECA&D rain gauge data

Starting point of the merging algorithm is that radar and rain gauge data are used from the same 1-h interval, instead of com-
puting adjustment factor fields based on preceding time intervals, such as in Park et al. (2019). Since the measurement interval
of daily rain gauge accumulations varies across Europe, a daily adjustment factor field would not be entirely representative.
And since the aim is to derive 1-h adjusted radar precipitation accumulations, ideally, 1-h adjustment factor fields would be
computed. To achieve this, the daily gauge accumulations are disaggregated to 1-h accumulations employing the 1-h and 24-h
radar accumulations from the previous processing step. It is assumed that the gauges observed precipitation only during the
intervals for which radar data were available. So in case of missing radar data, at most 4 hours per 24-h interval, the daily gauge
precipitation is only distributed over the remaining (at least 20) hours. Next, to decrease computation time of the radar-gauge
merging, only 1-h radar-gauge pairs for which gauge precipitation exceeds 0.25 mm are used for merging. This is expected to
have limited effect on the quality of the merged dataset.

The radar-gauge merging algorithm is based on Barnes' Objective Analysis (Barnes, 1964) but has been extended to make
it robust in case of sparse gauge network densities for short durations (1 h), when spatial precipitation variability is often
large. A spatial adjustment factor field $F_c$ is computed from the distance-weighted interpolation of the raw radar precipitation
accumulations ($S_{w,r}$) and the interpolation of the corresponding gauge precipitation accumulations ($S_{w,g}$), implying that it is
computed for each radar grid cell, which has position $(x, y)$:

$$
F_c(x,y) = \begin{cases} \frac{S_{w,r}}{S_{w,g}} & \text{if} \quad S_{w,r} > T \bigvee S_{w,g} > T \\ \frac{T}{S_{w,g}} & \text{if} \quad S_{w,r} \leq T \bigvee S_{w,g} > T \\ \frac{S_{w,r}}{T} & \text{if} \quad S_{w,r} > T \bigvee S_{w,g} \leq T \\ 1 & \text{if} \quad S_{w,r} \leq T \bigvee S_{w,g} \leq T, \end{cases}
\tag{1}
$$

with $T$ a threshold value of 0.25 mm. $S_{w,X}$, with X an indicator being g (gauge) or r (radar), is defined as:

$$
S_{w,X} = \sum_{n=1}^{N_P} w_n R_{X,n},
\tag{2}
$$

which is computed over $N_P$ radar-gauge pairs. $R_{r,n}$ and $R_{g,n}$ are the precipitation accumulations at the gauge location $n$ for
radar and gauge, respectively. In case the value of $S_{w,X}$ is below $T$, it is set equal to $T$. This is done to prevent outliers in
the gauge-adjusted radar precipitation accumulations. The weighting function $w_n$ depends on the distance of a grid cell to the
gauge location:

$$
w_n = \frac{G_w(n, r_s) + v \cdot G_w(n, r_l)}{1 + v},
\tag{3}
$$

where $G_w(n, r_d)$ is a Gaussian function:

$$
G_w(n, r_d) = \begin{cases} \frac{\exp\left(-4 \frac{(x-x_n)^2 + (y-y_n)^2}{r_d^2}\right) - \exp(-4)}{1 - \exp(-4)} & \text{if} \quad (x-x_n)^2 + (y-y_n)^2 \leq r_d \\ 0 & \text{if} \quad (x-x_n)^2 + (y-y_n)^2 > r_d, \end{cases}
\tag{4}
$$





and $(x_n, y_n)$ the position of gauge $n$. Equation 3 employs two Gaussian functions, each with its own characteristic distance $r_d$

for which the influence of a gauge is reduced to 0. The shorter range, $r_s$, results in a local adjustment in the neighbourhood of gauges. The value of $r_s$ is taken as the range of an isotropic spherical variogram model, which has been expressed as a function of day of year (DOY) and duration (1–24 h) using a 30-year rain gauge dataset from the Netherlands from the period 1979–2009 (Van de Beek et al., 2012). When distances from a grid cell to gauges are longer than $r_s$, this short range component does not contribute to the adjustment. The longer range, $r_l$, set at 500 km, is used to also adjust when gauge network density is

sparse and the nearest gauges are far away (e.g., over sea). The value of $v$ controls the contribution of this long range component with respect to the short range component. Apart from the actual adjustment in which all selected gauges are used, leave-one-out-statistics (LOOS) are also provided. These statistics are computed based on adjusted radar precipitation accumulations computed for a given gauge location without using that particular gauge in the adjustment. This is repeated for each gauge location, thus allowing for an independent verification of the gauge-adjusted radar dataset.

The algorithm can also run two adjustments consecutively with different settings, while still providing LOOS. Here, first an adjustment is performed with $v = 100000$, implying a local mean-field bias adjustment taking into account all gauges within a radius of 500 km. The short range component does not play a role then. This helps to remove systematic underestimations as much as possible in regions with low gauge network densities. Tests indicated that underestimations could not be effectively removed when also the short range component contributed. Next, $v$ is set to 0, implying that only a local spatial adjustment

is performed on top of the already mean-field bias adjusted precipitation estimates. The adjustment factor fields from both adjustments are combined into one 1-h spatial adjustment factor field. The 1-h radar precipitation composite is divided by this adjustment factor field to obtain the 1-h adjusted radar precipitation composite (EURADCLIM). Figure 4i–k illustrates the adjustment by showing an unadjusted radar composite, the adjusted radar composite and the adjustment factor field. The effect of the long and short range components is visible in the adjustment factor field.

## 3.5   Evaluation

The radar precipitation accumulations are evaluated against rain gauge accumulations by means of scatter density plots, maps with station-based spatial verification, comparison of annual precipitation maps to those based on gridded rain gauge data (E-OBS), and statistical metrics. In addition, maps of the mean hourly precipitation and the relative frequency of exceeding a threshold value of 1-h precipitation are compared between different processing steps. Statistical metrics used for evaluation

are the relative bias of radar precipitation accumulations compared to the corresponding gauge precipitation accumulations, the residual standard deviation divided by the mean gauge precipitation accumulation (i.e., the coefficient of variation, CV), the Pearson correlation coefficient ($\rho$) and the mean absolute error (MAE). Here, a residual is defined as the radar precipitation accumulation minus the gauge precipitation accumulation. Results are provided for all radar-gauge pairs as well as for the subset where the gauge exceeds a threshold of 1.0, 10.0, and 20.0 mm day$^{-1}$. Note that representativeness errors can be

significant when comparing radar and gauge accumulations (Kitchen and Blackall, 1992), especially for shorter durations, such as 1 h, and in case of larger differences in measurement volumes. The grid cell size of 4 km$^2$ is relatively large. Radars





measure aloft, and rain gauges measure at the Earth's surface, but only over a small area. Hence, differences between radar and gauge accumulations can be partly attributed to representativeness errors.

## 4 Results

The radar precipitation datasets are assessed by first systematically evaluating the influence of the clutter removal algorithms on QPE, followed by an evaluation of the performance of the EURADCLIM precipitation estimates. EURADCLIM's pan-European precipitation climatologies are shown and EURADCLIM's limitations are discussed. Finally, extreme precipitation events derived from EURADCLIM are presented.

### 4.1 Evaluation of clutter removal algorithms

Maps of mean hourly precipitation over the period 2013–2020 (Fig. 5a–b) show that the Gabella clutter filter removes and reduces many non-meteorological echoes, e.g., sea clutter at the North Sea, rings around Denmark, a radial pattern around Estonia, and spokes caused by interference over Slovakia. Also areas belonging to the highest precipitation class become smaller in Romania and southern France. Additional reductions are less pronounced when the cloud type mask is applied (Fig. 5b–c). A clear reduction in interferences in eastern Spain stands out and again areas falling in the highest precipitation class 270 generally become smaller. The static clutter mask is effective in (strongly) reducing many of the remaining non-meteorological echoes, but note that for some areas values may still be too high (Fig. 5c–d). And the area in Europe known for the highest annual precipitation, the coastal area of Norway, also shows a strong reduction, which may point to unwarranted classification of non-meteorological echoes.

From the maps of relative frequency of 1-h precipitation exceeding 0.1 mm (Fig. 5e–f) and 5 mm (Fig. 5i–j) it can be 275 concluded that the Gabella filter successfully removes non-meteorological echoes, especially sea clutter and suspicious noisy areas above land. The circles and radial patterns found for mean precipitation are not apparent, indicating that these do not occur frequently. For the 0.1 mm threshold, application of the cloud type mask clearly reduces the impact of interference above Spain and some areas belonging to the highest precipitation class become smaller, especially in eastern Europe and south of the French Mediterranean coast. Apparent is that larger areas above Germany and Spain now belong to a lower 280 frequency class, which could point to unwarranted classification of non-meteorological echoes. The cloud type mask hardly adds value in removing non-meteorological echoes for the 5 mm threshold (Fig. 5j–k). The static clutter mask successfully removes non-meteorological echoes for the 0.1 mm threshold (Fig. 5g–h) in eastern Europe and for the 5 mm threshold (Fig. 5k–l) remaining interferences are successfully removed over all of Europe. The fraction in mean rainfall and the fraction in relative frequencies of these datasets, which have undergone additional clutter removal, with respect to the OPERA dataset are 285 presented in Appendix A1,

Conclusion is that the algorithms remove many and also severe non-meteorological echoes. To evaluate the unwarranted removal of precipitation echoes, daily radar accumulations are compared to daily gauge accumulations in Table 1. This independent evaluation for different threshold values, shows that differences in the value of MAE between the four datasets

**Figure 5.** Comparison between 1-h precipitation accumulations from four datasets to study the effect of non-meteorological echo removal algorithms over the period 2013–2020. Results are only shown for grid cells with a minimum availability of 83.3%. (a)–(d) Maps of mean hourly precipitation, and (e)–(l) maps of relative frequency of exceedance of 0.1 and 5 mm in an hour. These are obtained by dividing by the number of available values for each individual radar grid cell (i.e., not equal to "nodata" or not missing). This implies that periods with missing radar data are not taken into account and the number of intervals that is used will vary in space. Maps made with Natural Earth. Free vector and raster map data ©naturalearthdata.com.

are small, the value of CV usually decreases and the value of $\rho$ usually increases after each additional processing step. The
strongest improvement is found for the Gabella clutter filter and the static clutter filter. The relative bias in the mean daily
precipitation becomes much more negative, though. The non-meteorological echoes may have compensated for the large un-
derestimation that is typical for mid-to-high latitude radar-only precipitation estimation (Overeem et al., 2009b). Because of





the general improvement for other statistical metrics, it is concluded that the clutter removal algorithms are effective and remove few precipitation echoes. Concluding, the radar dataset that will be used for merging with rain gauge data has improved

considerably over the dataset where no additional filtering is applied, in terms of the correlation and CV.

As a final check of the effectiveness of the clutter filter, daily radar accumulations are selected when the daily gauge accumulation is 0 mm. This results in average daily accumulations of 0.5 mm when no filtering is applied and this decreases to 0.3 mm for the Gabella filter and to 0.2 mm when also the cloud type mask is applied. It remains 0.2 mm when also the static clutter filter is applied. This confirms the effectiveness of the algorithms to further remove non-meteorological echoes.

## 4.2 Evaluation of EURADCLIM

In Table 1, a dependent verification of daily radar precipitation accumulations against rain gauges is performed to quantify the influence of the adjustment employing gauge data (i.e., with respect to the "Gabella + CTM + static filter" dataset). Results are presented for all values and for those where gauges exceed specific thresholds. Note that all selected values are used to compute the statistical metrics. A number of conclusions can be drawn from Table 1: 1) the severe average underestimation

of precipitation of ~45% turns into an overestimation of ~11%; 2) for daily gauge precipitation above 1 mm the relative bias is near zero; 3) the large underestimation for very heavy precipitation days (over 20 mm per day) is reduced from $\sim 65\%$ to about $\sim 10\%$; 4) the values for the correlation coefficient strongly increases from 0.59 to 0.89 based on all precipitation events; 5) the values for CV strongly decrease with a factor between 1.3 and 1.7; 5) the values for MAE decreases to values that are between 1.7 and 3.2 times smaller. The scatter density plots provide a more complete overview of the improvement and show

the much better alignment along the 1:1 line compared to the unadjusted dataset for daily gauge precipitation above 1 mm (Fig. 6a–b). The quality of daily precipitation accumulations is higher in summer than in winter (Fig. 6c–d). Although differences in the values of statistical metrics are relatively small, the spread in the scatter density plot for summer is clearly lower than in winter.

For hourly precipitation, both an independent verification via leave-one-out-statistics and a dependent verification are per-

formed. The scatter density plots show the radar-gauge pairs for 1-h gauge precipitation larger than 0.25 mm (Fig. 6e–g). The remaining underestimation becomes small and the value for the coefficient of determination ($\rho^2$) increases with respect to the unadjusted radar dataset, especially for the dependent verification. For EURADCLIM, the value for CV increases for the independent verification. The second lowest count class becomes rather wide above the 1:1 line. For the dependent verification the scatter density plot is much better aligned to the 1:1 line and the value for CV becomes much lower compared to the unadjusted

dataset. Conclusion is that the quality of the EURADCLIM dataset is good and much better than the dataset that has undergone the clutter filtering but no gauge adjustment. Note that the verification via LOOS is not entirely independent, because the 1-h and 24-h radar accumulations have been employed to disaggregate the daily gauge accumulations to hourly accumulations.

Next, a spatial verification per rain gauge location is performed for the radar dataset before gauge adjustment and for EURADCLIM, again for an independent (LOOS) and a dependent verification (Fig. 7). The quality of the radar composite for

the unadjusted radar dataset displays quite some variability and there seems some connection with areas further away from the nearest radar (Fig. 2c). Also different environmental conditions, e.g., beam-blockage due to mountainous terrain, can play a



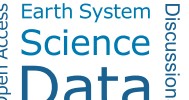

**Table 1.** Evaluation of radar daily precipitation accumulations against the ECA&D rain gauge network over the period 2013–2020 at their default measurement interval for all radar-gauge pairs and for those above a threshold value. The mean daily precipitation and the threshold value are based on the gauge data.

| Threshold value (mm) | Mean daily precipitation | Rel. bias (%) | Correlation | MAE (mm) | CV | $n$ |
|---|---|---|---|---|---|---|
| No filtering: | | | | | | |
| | 2.41 | -25.3 | 0.11 | 1.85 | 8.40 | 18946028 |
| 1.0 | 7.41 | -42.7 | 0.17 | 4.50 | 2.44 | 5970041 |
| 10.0 | 19.52 | -55.3 | 0.22 | 11.88 | 0.85 | 1299677 |
| 20.0 | 32.05 | -61.4 | 0.25 | 20.69 | 0.56 | 401277 |
| Gabella clutter filter: | | | | | | |
| | 2.41 | -38.8 | 0.19 | 1.61 | 5.06 | 18930248 |
| 1.0 | 7.41 | -47.9 | 0.42 | 4.34 | 1.15 | 5964987 |
| 10.0 | 19.52 | -57.6 | 0.35 | 11.91 | 0.63 | 1299043 |
| 20.0 | 32.05 | -63.0 | 0.29 | 20.86 | 0.50 | 401049 |
| Gabella + CTM clutter filter: | | | | | | |
| | 2.41 | -42.0 | 0.21 | 1.58 | 4.47 | 18936402 |
| 1.0 | 7.41 | -49.5 | 0.45 | 4.40 | 1.11 | 5967504 |
| 10.0 | 19.52 | -58.5 | 0.35 | 12.06 | 0.62 | 1299285 |
| 20.0 | 32.05 | -63.7 | 0.29 | 21.07 | 0.50 | 401091 |
| Gabella + CTM + static clutter filter: | | | | | | |
| | 2.41 | -44.9 | 0.59 | 1.55 | 2.01 | 18930115 |
| 1.0 | 7.41 | -51.0 | 0.59 | 4.39 | 0.96 | 5964938 |
| 10.0 | 19.52 | -59.9 | 0.41 | 12.16 | 0.57 | 1299042 |
| 20.0 | 32.05 | -65.2 | 0.32 | 21.30 | 0.47 | 401049 |
| EURADCLIM: | | | | | | |
| | 2.41 | 10.8 | 0.89 | 0.89 | 1.18 | 18929782 |
| 1.0 | 7.41 | -0.1 | 0.88 | 1.88 | 0.58 | 5964696 |
| 10.0 | 19.52 | -7.1 | 0.78 | 4.03 | 0.42 | 1298998 |
| 20.0 | 32.05 | -10.2 | 0.73 | 6.75 | 0.36 | 401047 |

role. The values for $\rho$, CV and relative bias in the mean strongly improve for the EURADCLIM dataset (dependent verification; Fig. 7c,f,i) with respect to the unadjusted radar dataset. In addition, the spatial variability in performance becomes much smaller. However, for the independent verification for regions with low gauge network densities, the value for $\rho$ sometimes decreases, the value for CV often increases and the underestimation becomes either less severe or turns into large overestimation


(Fig. 7b,e,h) with respect to the unadjusted radar dataset. Note that the dependent verification shows the actual performance of EURADCLIM at those locations, and that the independent verification is meant to give an impression of the quality between those locations. In reality, results are expected to be better than found for the independent verification, because the distance to the nearest gauge will be much shorter. After all, the gauges that are left out for LOOS have been used in the final EURADCLIM

dataset. Finally, for some gauge locations large differences are found. For instance, for two nearby stations in Poland a large overestimation and a large underestimation are found. Also the values for CV are high. This may point to erroneous rain gauge data. For other regions, radar beam-blockage could play a role.

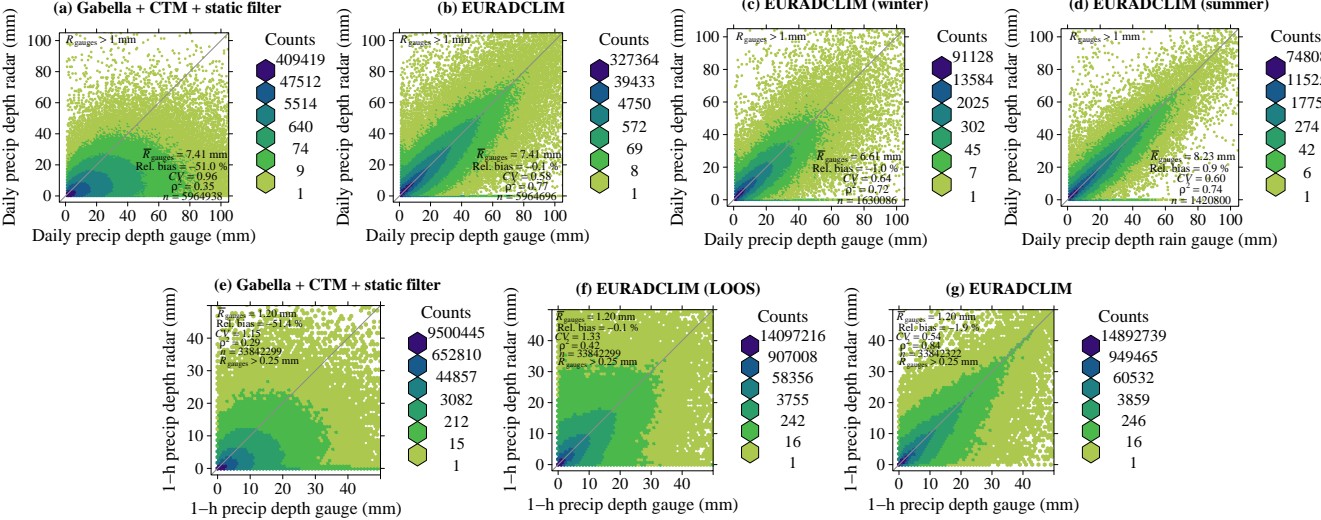

**Figure 6.** Scatter density plots of (a)–(d) daily and (e)–(g) 1-h radar precipitation accumulations against rain gauges over the period 2013–2020. For daily accumulations the gauge accumulations at their default measurement interval are employed, whereas for 1-h accumulations the disaggregated clock-hourly gauge precipitation accumulations are employed. Results are shown for the unadjusted dataset that has undergone all clutter filtering steps and for the gauge-adjusted EURADCLIM dataset. For EURADCLIM, independent verification is done via leave-one-out-statistics (LOOS), when indication. Otherwise, the verification is dependent.



**Figure 7.** Spatial verification of 1-h precipitation accumulations against the disaggregated clock-hourly gauge precipitation accumulations over the period 2013–2020. (a, d, g) For the unadjusted radar dataset that has undergone all clutter filtering steps. Results from the gauge-adjusted EURADCLIM dataset are shown for (b, e, h) an independent verification employing leave-one-out-statistics (LOOS) and for (c, f, i) a dependent verification. Maps made with Natural Earth. Free vector and raster map data ©naturalearthdata.com.



## 4.3 EURADCLIM radar precipitation climatologies and its limitations

Despite the efforts by NMHS and OPERA to remove non-meteorological echoes, and the three additional clutter removal
algorithms employed to derive EURADCLIM, still non-meteorological echoes can be persistent for some areas (Fig. 5h). Two
cases with strong artefacts are shown in Fig. 8 for the original OPERA surface rain rates. For the first case, the entire radar
domain of a Spanish radar has very high rates, which seems to be caused by a constant signal source that lasted the first 7 hours
of 29 April 2018 (Fig. 8a). Since this occurs over an entire radar domain, the algorithms can only partly remove and reduce
these non-meteorological echoes (unless entirely cloud-free or only semi-transparent clouds). This is much reduced by the
gauge adjustment, although still 24-h precipitation accumulations of more than 50 mm are found in EURADCLIM (Fig. 8b).
For the second case (Fig. 8c) with strong artefacts from a French radar, the 1-h precipitation accumulations are substantially
reduced by the EURADCLIM algorithms (Fig. 8d). Again, this reduction is primarily caused by the gauge adjustment.

Now the quality of EURADCLIM has been quantified, precipitation climatologies are derived to show its potential (Fig.
9). A map of mean hourly precipitation over the period 2013–2020 is shown in Fig. 9a. There are still some signatures of
beam-blockage, probably caused by obstacles near radar sites, and of non-meteorological echoes, such as interference. The
highest precipitation values are found in the coastal areas of Norway and in some mountainous areas (e.g., the Alps, Bosnia
and Herzegovina, Croatia, Ireland, Norway, Scotland, and Wales). The seasonality of mean hourly precipitation is visualized
in Fig. 9b–e. In Western Europe, precipitation is highest in autumn and winter. The Mediterranean areas are typically dry in
winter and summer and relatively wet in spring and autumn. Apparent is the high seasonal precipitation in the Alps in summer.
Relative frequencies of exceeding 0.1 mm are displayed in Fig. 9f, revealing that 1-h precipitation is most frequent in Denmark,
Ireland, large parts of the United Kingdom, parts of Norway, and a patchy band from France to Switzerland and Germany to
parts of Eastern Europe. The southern part of Spain has the lowest frequency of 1-h precipitation exceeding 0.1 mm. The
relative frequency of exceeding 5 mm (Fig. 9g) shows that more extreme precipitation occurs most often in parts of southern
Europe and parts of Ireland and the United Kingdom. It is difficult to tell to what extent non-meteorological echoes play a role
here. For instance, some of the localized areas with high frequencies of exceeding 5 mm may be related to obstacles near radar
sites. This becomes even more apparent for downpours of more than 25 mm in 1 hour (Fig. 9h), where large areas are found
with at least 13 occurrences in 8 years, sometimes present over a large part of a radar domain. These large values at the edge
of radar coverage are suspicious, especially when no nearby rain gauges are available, such as east of the island of Corsica
(France).

Outliers, such as presented in Fig. 8 and visible in the frequency of downpours (Fig. 9h), limit the applicability of EURAD-
CLIM at the grid cell scale, especially for use in extreme value modelling (e.g. Frederick et al., 1977; Durrans et al., 2002;
Allen and DeGaetano, 2005; Overeem et al., 2009a, 2010; Marra and Morin, 2015; McGraw et al., 2019). One way to deal with
this is to apply an algorithm that does not take into account entire composites that have severe artefacts over larger areas. These
could be automatically identified by radar-gauge comparison, possibly followed by a visual inspection and selection. Another
approach would be to set grid cells above a certain threshold to a lower (e.g., interpolated) value. Or always discard such a grid
cell if it experienced such a high value at least once during the period 2013–2020. A drawback of this approach is that such a

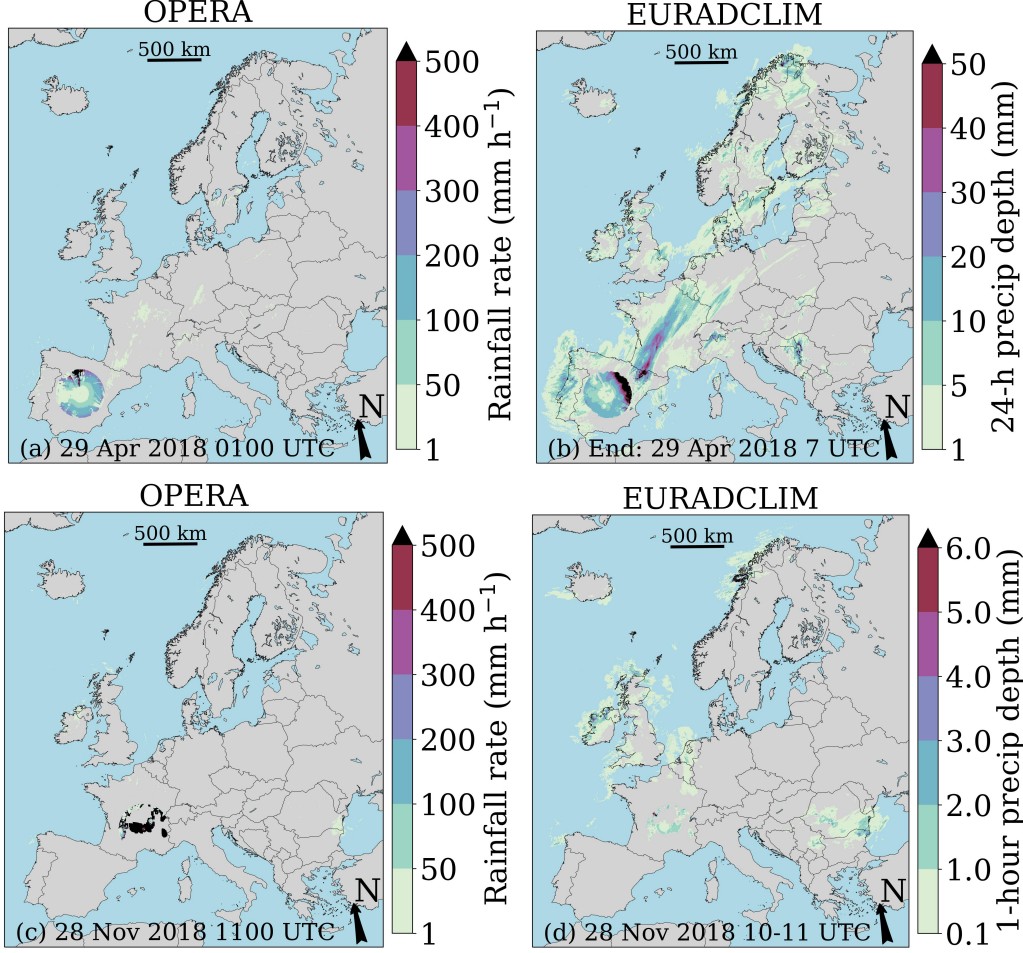

**Figure 8.** Illustration of remaining non-meteorological echoes in radar data for the original instantaneous OPERA rain rates and the corresponding 24-h or 1-h precipitation accumulations from EURADCLIM. (a)–(b) "Extreme precipitation" in central Spain on 29 April 2018 0100 UTC (rain rate) and from 28 April 2018 0700 UTC – 29 April 2018 0700 UTC (24-h precipitation accumulation). The gauge adjustment helps to lower the accumulations in EURADCLIM. (c–d) An extreme case with more than 500 mm h$^{-1}$ over large parts of a French radar domain on 28 November 2018 for 1100 UTC. Clutter removal algorithms hardly help, but the gauge adjustment substantially reduces the 1-h precipitation accumulations from EURADCLIM from 1000-1100 UTC. Maps made with Natural Earth. Free vector and raster map data ©naturalearthdata.com.

threshold is a bit arbitrary and that it influences statistics of extreme precipitation. A more sophisticated approach would be to remove more non-meteorological echoes by applying a satellite cloud mask that employs cloud optical thickness, i.e., also in case of thicker non-precipitating clouds. Finally, in the future, MSG will be replaced by Meteosat Third Generation that allows

for a more local correction due to higher spatial and temporal resolution, provided that parallax effects are accounted for.



**Figure 9.** (a)–(e) Maps of mean hourly precipitation over the entire period and per season over the period 2013–2020 (winter: December, January, February; spring: March, April, May; summer: June, July, August; autumn: September, October, November). (f)–(g) Relative frequency of exceedance of 0.1 and 5 mm in an hour over the period 2013–2020, respectively. (h) Frequency of exceedance of 25 mm in an hour, a downpour, over the period 2013–2020. Relative frequencies of exceedance ((f)–(g)) are obtained by dividing by the number of available values for each individual radar grid cell (i.e., not equal to "nodata" or not missing). This implies that periods with missing radar data are not taken into account and the number of intervals that is used will vary in space. For (a)–(g) results are only shown for grid cells with a minimum availability of 83.3%. Maps made with Natural Earth. Free vector and raster map data ©naturalearthdata.com.





An additional comparison is presented in Appendix A2, where the eight EURADCLIM annual precipitation accumulations are compared to those from the gridded rain gauge dataset E-OBS. Generally, precipitation patterns agree, but many local differences can be found. At far range from radars and rain gauges, a decrease in annual precipitation is found (e.g., above sea). Artefacts in radar accumulations, especially spokes caused by interference, result in overestimation in some regions. The

E-OBS data are based on all days, whereas EURADCLIM has some missing data, especially the first ∼three weeks of 2013 do not have data. Note that E-OBS has a grid of 0.1°, which is much coarser than the 2-km radar grid. More importantly, the underlying gauge network density can be sparse, whereas EURADCLIM provides full coverage. Hence, differences between EURADCLIM and E-OBS are expected and Appendix A2 is meant as an illustration and sanity check for both datasets.

### 4.4 EURADCLIM extreme precipitation events

Figure 10 shows EURADCLIM's precipitation estimates for three precipitation events: a wide-spread event across Europe showing the associated precipitation pattern of an extratropical cyclone with locally more than 60 mm in 24 hours (Fig. 10a). An extreme event in eastern Europe with at least 120 mm in 24 hours (Fig. 10b). And a very extreme event in southern France with at least 350 mm locally in 24 hours, of which at least 80 mm in the last hour (Fig. 10c–d). These events illustrate that EURADCLIM can capture extreme precipitation events across Europe, which can especially be valuable for regions where

climatological radar precipitation datasets do not exist or are not open, and where the affected area spans multiple countries.

### 5 Conclusions

We presented a climatological gauge-adjusted radar dataset of 1-h and 24-h precipitation accumulations, EURADCLIM, covering a large part of Europe over the period 2013–2020. Clearly, EURADCLIM will not outperform national (climatological) radar precipitation datasets (e.g. Overeem et al., 2009b; Goudenhoofdt and Delobbe, 2016; Winterrath et al., 2018; Saltikoff

et al., 2019a; KNMI, 2022). The spatiotemporal resolution of (the underlying) composites will often be higher, e.g., 5 min or 10 min instead of 15 min and 1 km$^2$ instead of 4 km$^2$. In addition, they may have undergone additional processing (e.g., based on 3D radar data) or may use more rain gauge data. These are expected to increase the accuracy of QPE with respect to EURADCLIM. However, such national datasets often do not exist or are not freely available for research or other purposes. Moreover, EURADCLIM allows users to use a common dataset for a large part of Europe, instead of using different datasets

from multiple countries. EURADCLIM will also benefit from possible future improvements in the OPERA surface rain rate product.

Apart from the evaluation of EURADCLIM, the performance of the OPERA precipitation product and three clutter removal algorithms were evaluated over an 8-year period. Some of the processing steps could also be applied in (near) real-time and would help to further improve the OPERA precipitation products. As is shown, the Gabella clutter filter would clearly decrease

non-meteorological echoes in the OPERA product and would be directly applicable. OPERA already uses a satellite cloud mask. The static clutter mask could be applied if the annual precipitation accumulations from the previous year would be used. When sub-daily near real-time gauge accumulations would become available, this would pave the way for merging with



**Figure 10.** Three precipitation events that led to flooding. (a) A widespread event over Europe from 30 May 2013 1400 UTC – 31 May 2013 1400 UTC (24-h precipitation accumulation), which is also presented as Supplement (Movie S1). (b) An event in eastern Europe from 14 May 2014 0000 UTC – 15 May 2014 0000 UTC (1-h precipitation accumulation), and (c)–(d) a very extreme event north of Nice, southern France from 1 October 2020 1400 UTC – 2 October 2020 1400 UTC (24-h precipitation accumulation) and its last hour (1-h precipitation accumulation). (a) Map made with Natural Earth. Free vector and raster map data ©naturalearthdata.com. (b)–(d) map data ©OpenStreetMap contributors 2022. Distributed under the Open Data Commons Open Database License (ODbL) v1.0.



OPERA radar accumulations in near real-time, e.g., by merging data from the last clock-hour. We think this would be a useful improvement of the product developed by Park et al. (2019). The correction for motion of the precipitation field from Park
et al. (2019) could be a valuable addition to EURADCLIM. Moreover, in line with Park et al. (2019), the temporal resolution of EURADCLIM could be increased to 15 min. Finally, the period of the evaluation by Park et al. (2019) of OPERA-based QPE as a function of time, could be extended, and this daily monitoring could also be applied to EURADCLIM.

This first version of EURADCLIM should be seen as a starting point. Despite the different algorithms to remove non-meteorological echoes applied by NMHS, OPERA and in EURADCLIM, these echoes do still pose a problem. We claim that
precipitation climatologies derived from EURADCLIM have a reasonable accuracy and that extreme events can be captured at a much higher spatiotemporal resolution, but that EURADCLIM is not directly suited yet for extreme value modelling at the grid cell scale, which requires records of the most extreme events, such as annual maxima. EURADCLIM may already be suitable for extreme value modelling in case of longer durations or larger area sizes than the grid cell scale, such as larger hydrological catchments. Then non-meteorological echoes may average out. QPE could be further improved, especially in
areas with sparse rain gauge network density or far away from weather radars. Some regions are far away from rain gauges, making the merging algorithm less effective. In some cases no merging is even carried out because of the long distance to the nearest rain gauge (e.g., Iceland, Malta). To improve EURADCLIM, but also the OPERA near real-time products, we provide the following main recommendations:

– Start with the volumetric radar data and apply algorithms (Goudenhoofdt and Delobbe, 2016) such as fuzzy logic clut-
ter removal (Berenguer et al., 2006; Gourley et al., 2007; Crisologo et al., 2014; Krause, 2016; Overeem et al., 2020), attenuation correction (Carey et al., 2000; Testud et al., 2000; Vulpiani et al., 2012; Al-Sakka et al., 2013; Jacobi and Heistermann, 2016; Overeem et al., 2021) and vertical profile of reflectivity correction (Hazenberg et al., 2013). Especially the use of polarimetric variables would add value, also in the conversion to rain rates. Moreover, differences between weather radars, e.g., polarimetric or not, and climate would require careful local analyses and optimization of
parameters (e.g., fuzzy logic settings, precipitation retrieval relations for rain/snow).

– Obtain gauge data from more locations and update the rain gauge records that overlap with the (extended) EURADCLIM time span. Additional high quality data might come from, e.g., NMHS, water and river authorities, and other data-holding institutes. Instead of using only daily data, ideally also clock-hourly data would become available, which would avoid possible errors introduced by disaggregating daily precipitation. This is desirable given the often large spatial and
temporal variability in precipitation and in sources of error in radar QPE.

– Specifically, third party data could be employed after appropriate quality control (and retrieval) algorithms have been applied. This also requires efforts from NMHS to bring these from research to operations (Garcia-Marti et al., 2022). Examples of such third-party data sources are rain gauge data from personal weather stations (PWS) connected to the internet (De Vos et al., 2019; Graf et al., 2021) and commercial microwave link (CML) data (Messer et al., 2006;
Leijnse et al., 2007; Overeem et al., 2016; Graf et al., 2020, 2021). The merging of these kind of datasets with radar data is studied in the COST action OPENSENSE (https://opensenseaction.eu/). These data sources could potentially be





available in (near) real-time and would give a vast increase in the density of surface precipitation estimates. However, the increase over time of the number of available third party data, challenges the aim of EURADCLIM to provide a climatic perspective on hourly rainfall.

– Develop a radar-gauge merging algorithm that takes into account the local rain gauge network density. For the current method, the local mean-field bias adjustment followed by a spatial adjustment could then be replaced by a spatial adjustment where the value of the short range parameter varies seamlessly in space as a function of network density. In addition, the value of the short range parameter should be based on the local precipitation climatology instead on the precipitation climatology from the Netherlands. Also other adjustment methods could be evaluated (Goudenhoofdt and
Delobbe, 2009).

  – Add uncertainty information to radar and gauge data and use this in the radar-gauge merging. For instance, by using the OPERA quality index and additional information on the radar quality and the rain gauge network density. Alternatively, it could be studied whether a meaningful relationship can be established between the quality of EURADCLIM and the distance to the nearest gauge or radar. Then maps with distance to the nearest gauge or radar (Fig. 2c–d) could guide the
user to judge the suitability of using a certain grid cell or region of EURADCLIM for a given application. Ideally, such quality maps would be computed for each 1-h interval, taking into account the gauges actually used in the merging and the radar coverage and quality for that time interval. To facilitate this, we recommend to add the actual radar coordinates for each individual time interval to the OPERA products.

EURADCLIM's strategic value encompasses:

– Much better reference for validation of weather prediction model output (e.g., HARMONIE/ECMWF) (Van der Plas et al., 2017), Regional Climate Model simulations (Berg et al., 2019), satellite precipitation products (Skofronick-Jackson et al., 2018; Sun et al., 2018), and opportunistic sensing data (e.g., CML and PWS), which allows for improving their retrieval algorithms.

  – Better monitoring of (trends in) precipitation extremes and their spatial extent, except for the most extreme events at the
grid cell scale, such as annual maxima, due to remaining non-meteorological echoes. This facilitates better understanding of the drivers behind such events (e.g., the relation to dew point temperature, atmospheric circulation, diurnal cycle, clustering of showers) and climate attribution (Lochbihler et al., 2017; Lengfeld et al., 2019). Deriving a catalogue of extreme precipitation events becomes possible, as is done for Germany by Lengfeld et al. (2021). This is all highly relevant for anticipating future extremes in a changing climate.

– Better evaluation of extreme precipitation events and their impact (e.g., landslides, flooding). Specifically, use as input for hydrological models in order to improve these models, especially for flash flood forecasting.

We expect to rerun EURADCLIM once a year over the entire period, using all available ECA&D rain gauge data, and extend it with one year of data. This will result in a new version of this dataset. Hence, we invite NMHS or other institutes





to make all their rain gauge data available to ECA&D. We encourage to start using EURADCLIM and discover its value and
shortcomings for a variety of applications, and we appreciate any feedback on this. It is expected that EURADCLIM will be
of use to various (scientific) domains and applications and that future collaborations and funding will lead to improved next
versions of EURADCLIM.

## 6   Code availability

The following tools, written in programming language Python (version 3), are publicly available at the GitHub repository
EURADCLIM-tools (https://github.com/overeem11/EURADCLIM-tools), to process OPERA-based radar precipitation files:
a script to accumulate data, a script to perform climatological analyses (e.g., to compute the mean and the relative frequency of
exceedance), two scripts to visualize precipitation, one of them with an OpenStreetMap or Google Maps as background. This
helps end users to reproduce results from this study and to further explore and analyze the EURADCLIM dataset.

## 7   Data availability

The EURADCLIM 1-h and 24-h precipitation datasets are available from the KNMI Data Platform. The dataset of 1-h precipi-
tation is available at https://doi.org/10.21944/7ypj-wn68 (Overeem et al., 2022a). The dataset of 24-h precipitation is available
at https://doi.org/10.21944/1a54-gg96 (Overeem et al., 2022b). For each year a zip file is provided. The data are in UTC, where
the time in the unzipped filenames is the end time of observation in UTC.

*Video supplement.*   The supplemental video visualizes a widespread precipitation event on 30 and 31 May 2013 over Europe that led to
flooding. Supplement (Movie S1).



## Appendix A1: Effect of additional clutter removal algorithms on 1-h precipitation

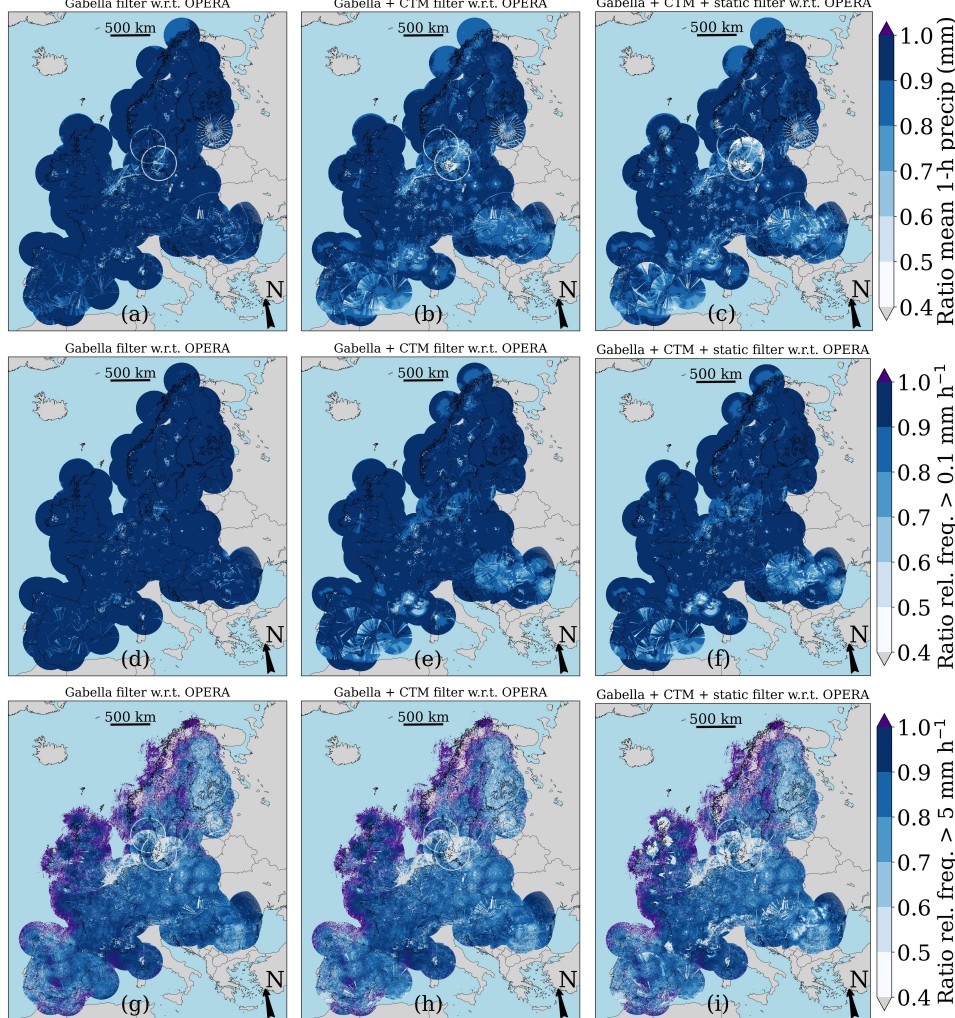

**Figure A1.** Comparison between 1-h precipitation accumulations from three datasets to study the effect of non-meteorological echo removal algorithms over the period 2013–2020. (a)–(c) Maps of the ratio of mean hourly precipitation with respect to the OPERA dataset, and (d)–(i) maps of the ratio of relative frequency of exceedance of 0.1 and 5 mm in an hour with respect to the OPERA dataset. The purple areas imply that the ratio is 1. The uncolored areas do not have data or have a relative frequency of 0 for one or both datasets. Map made with Natural Earth. Free vector and raster map data ©naturalearthdata.com.



## Appendix A2: Annual precipitation EURADCLIM versus E-OBS

**Figure A2.** Annual precipitation accumulations over the period 2013–2020 for EURADCLIM (4 km$^2$) and interpolated rain gauge observations (E-OBS version 25.0e; release April 2022; $0.1° \times 0.1°$, which is $\sim 11$ km in latitude and $\sim 4 - 9$ km in longitude, depending on the latitude). Coastlines are plotted in orange to ease the comparison between EURADCLIM and E-OBS, due to the coverage above open water by radars for which E-OBS does not provide precipitation estimates. Map made with Natural Earth. Free vector and raster map data ©naturalearthdata.com.





*Author contributions.* Author contribution is captured following the CRediT system. Conceptualization by AO, GS, HL. Data curation by AO, EB, GS, JFM. Formal Analysis by AO. Funding acquisition by AO, EP, HL, JFM. Investigation by AO, EB, GS. Methodology by AO, GS, HL, JFM. Project administration by AO. Software by AO, HL, JFM. Supervision by AO, HL. Validation by AO, EP. Visualization by AO. Writing – original draft preparation by AO, HL. Writing – review & editing by AO, EB, EP, GS, JFM, HL.

*Competing interests.* We have no competing interests.

*Acknowledgements.* We thank Dr. Martin Stengel (Deutscher Wetterdienst) for providing the CLAAS-2 satellite product and thank EUMET-SAT. We acknowledge the E-OBS dataset (Cornes et al., 2018) from the EU-FP6 project Uncertainties in Ensembles of Regional ReAnalyses (http://www.uerra.eu) and the Copernicus Climate Change Service, and the data providers in the ECA&D project (https://www.ecad.eu), as well as the NMHS who provided radar data to OPERA. We are grateful to Dr. Annakaisa van Lerber (Finnish Meteorological Institute, the OPERA program manager) and Willie McCairns MBA FRMetS (ECOMET, an economic interest grouping of the National Meteorological Services of the European Economic Area, Chief Executive) and Ms. Annelies Demolder (ECOMET) for their efforts to make the EURADCLIM dataset open. Project EURADCLIM was financed by KNMI's multi-annual strategic research programme (project number 2017.02). A website of the project can be found at https://www.knmi.nl/research/observations-data-technology/projects/euradclim-the-european-climatological-high-resolution-gauge-adjusted-radar-rainfall-dataset. All figures containing maps were made with Python package Cartopy (Met Office, 2010 - 2015).





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
