# Peer review of "EURADCLIM: The European climatological high-resolution gauge-adjusted radar precipitation dataset"

_Earth System Science Data, 2022_

## Referee Comment (RC2)

**Ref**: essd-2022-334_review

**Title**: EURADCLIM: The European climatological high-resolution gauge-adjusted radar precipitation dataset

**Authors**: Aart Overeem*, Else van den Besselaar, Gerard van der Schrier, Jan Fokke Meirink, Emiel van der Plas, and Hidde Leijnse

**General comments**

The data set is unique in the sense that the radar-rain gauge merging (hourly) is done at a European scale (over OPERA radar coverage) and is easily accessible. Sources of errors and limitations are discussed, and despite some limitations of the EURADCLIM dataset, this dataset can be useful in providing high-resolution (2 km, 1 hour) precipitation information.

However, the manuscript (and dataset) lacks clarity regarding its usage for further applications (e.g., missing indicators for erroneous EURADCLIM data, mainly literature-based ideas for potential applications, unclear definition of extreme cases). The authors created 1-hour and 24-hour rainfall accumulations with separate references. From the 1-h dataset, the 24-h dataset is generated in an hourly moving window and is 3 times larger in size than the 1-h dataset. The manuscript does not explain any special processing for the generation of this 24-h dataset nor its benefit. It is not clear why 24-h dataset should be produced separately with a separate reference.

Overall, the topic fits the scope of the journal's special issue, but the manuscript needs to be better clarified for its publication. Please see the comments below. Line numbers are indicated with "L"

Recommendation: Major

1. Section 4.3 and Fig.8: Two examples are presented to show the limitation of the EURADCLIM data (e.g., failing of clutter filtering effect). However, those radar echoes seem clearly a failure of radar processing for certain time steps (0045-0630 UTC which affected hourly EURADCLIM data from 0100-0700 that are more than 4 hours - please see also comment #11-, as briefly mentioned in L340 as well), which is unlikely "extreme precipitation" as stated in Figs. 8a and 8b. i) Have the authors cross-validated those events with other sources confirming the rainfall was extreme over the affected areas? ii) If the EURADCLIM aims to construct quality-checked past data (not in real-time adjustment) and those erroneous outputs are caused by at least a processing artefact (among other artefacts) that cannot be corrected even with the gauge adjustment for the limitation they pointed out, then the output analyses are expected how to deal with those time steps in their hourly merging and 24-hour accumulations or further applications. Instead, those results were explained in the context of the added value of the gauge adjustment (L347), which lacks discussion of the "limitation" of the EURADCLIM data. Although in L365-L372, the authors briefly mentioned a couple of ways to deal with those outlier cases, these do not seem to be applied to the current version of the EURADCLIM dataset. If an end-user is aware of such an outlier, the current version of the dataset will be unlikely used. Can the authors better propose or discuss what can help the users to use their climatological dataset?

2. Section 4.4: This section lacks a supporting explanation. How do they identify extreme events out of 8 years dataset? only by the radar 24 h accumulations? or a couple of flooding cases? For the presented extreme case analyses, were the outliers visually checked (as mentioned in section 4.2)? Over the flood-affected areas, have the authors checked that there were no rainfall estimates available produced (reanalyzed after the event) by regional and national products (that can justify L389-390)?

3. Unclear conclusion: (L423-L455). I am puzzled by their "recommendations" that seemed to target those who may replicate EURADCLIM. The improvement of EURADCLIM data will be made most likely by the authors, no? On the other hand, if the authors tried to promote their datasets to be used, I expect the recommendations in the context of the user perspective with some technical tips or examples, which perhaps they intended to address some with "strategic value" (L460-472). However, those "strategic values" are mostly speculated by addressing examples from the literature. One or two examples with detailed guidelines on the usage of the EURADCLIM data will be more useful (e.g., by improving section 4.4).

**Minor comments**

1. L43-L44: The method of Park et al. (2019) is rather based on a systematic bias adjustment, instead of merging. Hence, the following sentence can be revised (with bold) as "an operational **gauge-adjusted** OPERA-based radar rainfall product for the European Rainfall-InduCed Hazard Assessment (ERICHA) system. ~~employing data from ~1500 rain gauges.5~~ **6** hours for the European Flood Awareness System (EFAS)". Here, the number of rain gauges can be misleading because the daily bias map has been obtained from valid radar-rain gauge pairs, which may vary from different gauge sources, quality of both radar and gauge estimates, and days of rainfall over available gauge points. In any case, the algorithm can adapt to the use of more gauges.

2. L45-L46: "The use of this dataset is restricted to EFAS", This is not correct (e.g., the dataset and the adjustment algorithms have continuously been used in the framework of several research projects since 2017) and do not add any relevant information; so, please remove it.

3. L50-L51: Please remove "and data from far more rain gauges are available (~7700)". Perhaps the authors can mention that their dataset is built with different gauge sources from Park et al. (2019), but it is not necessary to point out the number of gauges unless adding an interesting research argument. The more gauges available, the better both methods would perform. In fact, the authors describe the gauge network used in the presented methods (in L104-L120), addressing the number of gauges deployed (~7700), which fits better here and shows the distribution clearly as well.

4. L56-L57: "Since it could also be applied in (near) real-time, its evaluation is also relevant for the existing gridded OPERA products". Can this be better clarified?

5. L76: "It is based on the raw single site radar data, which have undergone Doppler clutter filtering." Can this part be better explained?

6. L90-L95: Are the values of "nodata" and "undetect" directly from the OPERA 15-minute rain rate product? Or when calculating the 1-h precipitation accumulation, those values are assigned to the EURADCLIM data cell if not satisfying the full availability (similarly done as the OPERA data)? Here, can this full availability be better explained?

7. Fig. 2b, and L107: What does "Combined radar-gauge data availability (in %)" mean?

8. L103: Is it necessary to state "mid-June 2022" here? It is actually mentioned better in L115.

9. L121-L130: The authors comment that there are some uncertainties on the gauge aggregation time for the given ECA&D 24-h accumulation dataset and explain that "The end times of the observations display a large variability" in L127. Then, is such variability considered in the disaggregation of the 1-hour dataset applied in L199-L206?

10. Fig.3: For disaggregation, are the 24-h radar precipitation composites used?

11. L205: "at most 4 hours per 24-h interval", does this threshold of 4 hours come from the observed results? I am a bit puzzled, there were days with missing hours of more than 4 hours (as in the case of Fig. 8.a), and if this is the case, how would the correction(merging) be applied?

12. Fig. 5 a-d: Maps of mean hourly precipitation over 2013-2020. The mean values are so low, which is not so evident to see the effects over some areas. Is it meaningful to present in terms of mean 1-h precipitation to highlight the effect of clutter filters?

13. L271-L273: Although the authors state that there has been a strong reduction in the coastal area of Norway (5a-5d), it is not clear to see. Can it be explained better?

14. Table 1: Why the values of mean daily precipitation corresponding to 4 different filters are the same?

15. L289: I guess "rho" here is the correlation coefficient. (Please add the symbol in the table or explicitly state "correlation" next to the symbol in L289).

16. L297-L300: "This confirms the effectiveness of the algorithms to further remove non-meteorological echoes". Can this be better clarified? For no-rain only (0 mm), the effect of the static mask seems to be none based on daily accumulations on average.

17. L305: Regarding the results of an overestimation of 11%, please indicate explicitly EURADCLIM in the text.

18. L311: How do the authors separate between summer and winter cases?

19. Fig.6b-6d: Why there are some systematic "0" or near 0 values of EURADCLIM for all values of gauges?

20. L334-335: "After all, the gauges that…in the final EURADCLIM dataset". Can this be better clarified? Does it mean that those gauges are used in the evaluation of the EURADCLIM dataset? Or if these are a part of EURADCLIM, how does this justify (link to) the previous statement?

21. L337: "For other regions, radar beam-blockage could play a role." – Can this be explained better related to Fig.5 and Fig. 7 (e.g., Is it possible to identify those areas which indeed indicate the presented filter that does not seem to work based on the 8-year dataset? If so, in the comparison with gauges, those areas could have been excluded as well).

22. Fig.9: Why not use the same colour scale as those presented in Fig. 5? Can this be done in terms of daily accumulation?

23. L349-L350: Could it be better explained where those signatures are shown (in Fig. 9a or in Fig 9)? As mentioned in #22, can the results be better explained with respect to Fig. 5?

24. L359: "It is difficult to tell to what extent non-meteorological echoes play a role". It is possible that invalid(strange) radar data (similar to the presented case in Spain) are included in the 8-year statistics over Croatia, Kosovo, Slovakia and Corsica. Have the authors checked those areas in particular?

25. L407-L409: This conclusion needs better supporting materials or references; Because the presented method is based on neither sub-daily gauge data nor a real-time feed environment, it is not clear yet what aspects can be improved. For production in real-time, better improvements in L365- L375 (Major comment 1) are expected.
26. Code availability: It is good that the authors provide some routines, but this code will not be sufficient to "reproduce" the results and the end-users required a good knowledge of the OPERA data structure. So, rephrase or remove "This helps end users to reproduce results from this study and to further explore and analyze the EURADCLIM dataset."
27. Data availability: Hourly dataset is useful, however, if the 24-h precipitation datasets are simply generated by summing up hourly data in running windows, is there any specific reason to produce such big-size outputs with a separate reference?

---

## Author Comment (AC1)

Reply to comments by Anonymous Referee #1 (RC1)

**Our response to the comments by reviewer 1 is provided in bold font.**

The manuscript presents a new precipitation dataset that covers most parts of Europe and is based on the OPERA gridded radar dataset. The algorithms for filtering non-meteorological echoes are described and evaluated as well as the adjustment to gauge data. Limitations of the dataset are discussed and ways to improve the climatological precipitation dataset are given.

The manuscript fits in the scope of ESSD, it is well written and clearly structured. The described new European radar climatology is unique, of high interest and importance for the community and allows for a variety of applications and studies. I recommend publishing the manuscript after taking the following (minor) suggestions and comments into account:

**We thank the reviewer for recognizing the value of our dataset and of our manuscript. The constructive feedback is appreciated.**

- 3, L.83-85: Does that mean, that in case of 10 minute temporal resolution the 10-min file from 10:10 UTC is used for the 10:15 UTC composite?

**Yes, we will clarify this by replacing "The measurement frequency of the radars is 5 min or 10 min, and data from the last time stamp are used in the composite product, e.g., the 5-min file from 10:15 UTC for the 10:15 UTC OPERA composite product." by "The measurement frequency of the radars is 5 min or 10 min, and data from the last time stamp are used in the composite product. For instance, the 5-min file from 10:15 UTC and the 10-minute file from 10:10 UTC for the 10:15 UTC OPERA composite product."**

- 11, Eq.1: Why is $S_{w,g}$ set to the value of T (0.25 mm) in case it is lower than T? In line 207 the authors say that 1-h radar-gauge pairs are only used for merging if the gauge precipitation exceeds 0.25 mm. Wouldn't that mean that no factors should be computed in case $S_{w,g}$ is lower than T?

**Indeed, only 1-h radar-gauge pairs are used for merging if the gauge precipitation exceeds 0.25 mm. $S_{w,g}$ is the distance-weighted interpolation of the gauge precipitation accumulations, i.e., the summation, over all radar-gauge pairs, of the gauge value times its weight, which depends on the distance from a radar grid cell to the radar-gauge pair. Hence, only gauge values larger than 0.25 contribute to $S_{w,g}$, and lower gauge values are not used at all in the adjustment. So the value of T (0.25 mm) is not related to the use of gauge values exceeding 0.25 mm, and is only used to prevent outliers in the gauge-adjusted radar accumulations.**

- 12, L. 235-241: An example of the adjustment fields for $v = 100000$ and $v = 0$ would be beneficial to understand the influence of the mean-field bias and the local spatial adjustment.

**This is actually already revealed in Figure 4 (k). We will add to the text (addition in italic font): The effect of the long and short range components is visible in the adjustment factor field, *where the large scale patterns belong to the long range component (v = 100000) and the local patterns (dots) show the influence of the short range component (v = 0) on top of the long range component.***

•20, Fig.8: It might be better to use the same colourbar for OPERA and EURADCLIM. Especially in the upper example the smaller range of precipitation values in the colourbar makes the EURADCLIM product look worse than the OPERA product. Maybe a logarithmic scale can help to compensate the different ranges of precipitation values.

**Note that Figure 8 shows instantaneous rainfall rates for OPERA and 1-h or 24-h accumulations for EURADCLIM. Hence, using the exact same colourbar is not possible. But we do agree that the colourbar can be chosen differently, to avoid that EURADCLIM looks worse than the OPERA product, whereas its quality is better. Therefore, we will change the colourbar of the rain rate in Figure 8 (a) and (b). The scale now ends at 140 mm/h instead of 500 mm/h:**

[Figure]

•22, Section 4.4: Have the authors compared their results to the corresponding national radar data sets? How similar are the extreme values in EURADCLIM and the national products?

**We did not make such a comparison, although the comparison to ECA&D data already provides such an assessment for rain gauge locations (Table 1). A comparison to national radar datasets is out of the scope of this study, but is definitely an interesting idea. Hence, we will add the following recommendation to the manuscript: "We also recommend to compare the EURADCLIM precipitation accumulations to those from national radar datasets, specifically to assess the performance of EURADCLIM to capture extreme precipitation."**

---

## Author Comment (AC2)

**Ref**: essd-2022-334_review

**Title**: EURADCLIM: The European climatological high-resolution gauge-adjusted radar precipitation dataset

**Authors**: Aart Overeem*, Else van den Besselaar, Gerard van der Schrier, Jan Fokke Meirink, Emiel van der Plas, and Hidde Leijnse

**General comments**

The data set is unique in the sense that the radar-rain gauge merging (hourly) is done at a European scale (over OPERA radar coverage) and is easily accessible. Sources of errors and limitations are discussed, and despite some limitations of the EURADCLIM dataset, this dataset can be useful in providing high-resolution (2 km, 1 hour) precipitation information.

However, the manuscript (and dataset) lacks clarity regarding its usage for further applications (e.g., missing indicators for erroneous EURADCLIM data, mainly literature-based ideas for potential applications, unclear definition of extreme cases). The authors created 1-hour and 24-hour rainfall accumulations with separate references. From the 1-h dataset, the 24-h dataset is generated in an hourly moving window and is 3 times larger in size than the 1-h dataset. The manuscript does not explain any special processing for the generation of this 24-h dataset nor its benefit. It is not clear why 24-h dataset should be produced separately with a separate reference.

Overall, the topic fits the scope of the journal's special issue, but the manuscript needs to be better clarified for its publication. Please see the comments below. Line numbers are indicated with "L"

**We thank the reviewer for recognizing the value of the EURADCLIM dataset and manuscript and for the detailed and constructive review. Below, we give attention to the reviewer comments on the usage for further applications. One general remark is that we think we provide a thorough evaluation of the EURADCLIM dataset, also giving attention to its limitations. We also believe that the EURADCLIM dataset in its current form clearly adds value to the existing pan-European datasets that are available.**

**The processing of the 24-h dataset is already explained in L93-96 in the original manuscript: "These 1-h precipitation accumulations are used to compute 24-h accumulations every clock-hour as well as annual accumulations. For each radar grid cell, a minimum data availability of underlying 1-h accumulations of 83.3% is demanded (i.e., at least 20 of 24 hours, or ~304.2 of 365 days). Grid cells with too low availability are set to the OPERA "nodata" value.".**

**We recognize that the reason for providing 24-h accumulations should be described. Reasons for providing 24-h accumulations is that these can be useful for, e.g., evaluation of extreme precipitation, evaluation of EURADCLIM by comparing to 24-h rain gauge accumulations (from ECA&D or other sources), and climatological analyses. Note that 24-h accumulations are a common measurement interval for (manual) rain gauge data. Daily data are for instance provided by the ECA&D portal as timeseries and by the associated gridded E-OBS dataset. Since the 24-h accumulations had already been generated for the manuscript, we decided to offer this as a service to users. This not only saves time for potential users, but also provides a common benchmark 24-h dataset, taking into account data availability and "nodata" values. We will add to the Introduction: "*Having 24-h accumulations can, for instance, be useful for evaluation of extreme precipitation events, evaluation of EURADCLIM against*

*(manual) rain gauge accumulations, and climatological analyses. Moreover, the ETCCDI Climate Indices require daily precipitation amounts for climate monitoring (https://www.wcrp-climate.org/etccdi). In addition, a drawback of the gridded E-OBS dataset is that the underlying rain gauges are aggregated over daily measurement intervals that are not homogeneous over Europe. The unique nature of EURADCLIM improves upon this aspect. Hence, the EURADCLIM dataset is not only available as 1-h, but also as 24-h accumulations."*

Recommendation: Major

1. Section 4.3 and Fig.8: Two examples are presented to show the limitation of the EURADCLIM data (e.g., failing of clutter filtering effect). However, those radar echoes seem clearly a failure of radar processing for certain time steps (0045-0630 UTC which affected hourly EURADCLIM data from 0100-0700 that are more than 4 hours - please see also comment #11-, as briefly mentioned in L340 as well), which is unlikely "extreme precipitation" as stated in Figs. 8a and 8b. i) Have the authors cross-validated those events with other sources confirming the rainfall was extreme over the affected areas? ii) If the EURADCLIM aims to construct quality-checked past data (not in real-time adjustment) and those erroneous outputs are caused by at least a processing artefact (among other artefacts) that cannot be corrected even with the gauge adjustment for the limitation they pointed out, then the output analyses are expected how to deal with those time steps in their hourly merging and 24-hour accumulations or further applications. Instead, those results were explained in the context of the added value of the gauge adjustment (L347), which lacks discussion of the "limitation" of the EURADCLIM data. Although in L365-L372, the authors briefly mentioned a couple of ways to deal with those outlier cases, these do not seem to be applied to the current version of the EURADCLIM dataset. If an end-user is aware of such an outlier, the current version of the dataset will be unlikely used. Can the authors better propose or discuss what can help the users to use their climatological dataset?

   **The examples indeed seem to illustrate a failure of radar processing, which can hence not be removed effectively employing postprocessing clutter removal algorithms. Calling them echoes caused by failure of radar processing is indeed more specific than calling them non-meteorological echoes. Hence, we replaced "Since this occurs over an entire radar domain, the algorithms can only partly remove and reduce these non-meteorological echoes" by "Since this occurs over an entire radar domain, the algorithms can only partly remove and reduce these *echoes caused by failures of radar processing*". We used parenthesis for extreme precipitation to denote that it is not extreme precipitation. We replaced "extreme precipitation" by "extreme values". i) Since the values are clearly artefacts there is no need for cross-validation. ii) We will add to Section 4.3: "*Still, Fig. 8b,d indicates that EURADCLIM sometimes contains substantial artefacts.*"**

   **In L365-L372, we suggest some pragmatic ways to deal with remaining outliers in EURADCLIM. We believe some of these could be applied by end users. We think EURADCLIM will be of value to end users, despite these outliers. Otherwise, end users would not be using (climatological) satellite precipitation products around the globe (indirect, relatively inaccurate), or gridded rain gauge datasets, such as E-OBS. Though gauge data will generally contain fewer outliers compared to EURADCLIM, they don't capture precipitation over the vast areas between rain gauges. This is exactly the added value EURADCLIM provides. Also note that it depends on the application end users have in mind whether outliers pose a (larger) problem.**

   **Note that we transformed L365-L372 to a list of recommendations for end**

users in the conclusions section, and moved L372-375 to the list of recommendations for NMHS, OPERA and EURADCLIM to improve OPERA-based precipitation products. We also made the two end user recommendations more specific. See our reply to comment #3.

2. Section 4.4: This section lacks a supporting explanation. How do they identify extreme events out of 8 years dataset? only by the radar 24 h accumulations? or a couple of flooding cases? For the presented extreme case analyses, were the outliers visually checked (**as** mentioned in section 4.2)? Over the flood-affected areas, have the authors checked that there were no rainfall estimates available produced (reanalyzed after the event) by regional and national products (that can justify L389-390)?

   **Section 4.4 is not meant to provide the most extreme events over the 8-year period, which we do not claim, but meant as an illustration of extreme events captured by EURADCLIM. These cases were selected by consulting a website on extreme flood events, already showing that severe precipitation occurred. We will add to Section 4.4 (additions in italic font): "Figure 10 shows EURADCLIM's precipitation estimates for three *(more) extreme* precipitation events *meant to illustrate EURADCLIM's potential*". Comparison with other radar-based precipitation products is out of the scope of this study.**

3. Unclear conclusion: (L423-L455). I am puzzled by their "recommendations" that seemed to target those who may replicate EURADCLIM. The improvement of EURADCLIM data will be made most likely by the authors, no? On the other hand, if the authors tried to promote their datasets to be used, I expect the recommendations in the context of the user perspective with some technical tips or examples, which perhaps they intended to address some with "strategic value" (L460-472). However, those "strategic values" are mostly speculated by addressing examples from the literature. One or two examples with detailed guidelines on the usage of the EURADCLIM data will be more useful (e.g., by improving section 4.4).

   **We find these recommendations important to emphasize that there is still much room for improving OPERA-based radar precipitation products in general, and not only EURADCLIM. Hence, the recommendations are also directly relevant for the next phase of OPERA, which is being planned. In the end, the improvement of EURADCLIM will be a joined effort by NMHS, OPERA and the EURADCLIM authors (or perhaps even in a collaboration in a new project). For instance, NMHS could perform better clutter removal employing dual-pol based algorithms and send these data to OPERA. The next phase of OPERA may apply methods to further improve quantitative precipitation estimates by applying additional algorithms to volumetric data. And EURADCLIM may gather more rain gauge data to be used in the merging. Moreover, the recommendations also reveal limitations in OPERA-based products, which can also be relevant for future funding and projects (limited gauge network densities, corrections not being applied to volumetric radar data). We acknowledge that technical tips or examples how to deal with limitations of EURADCLIM for specific applications, as indicated in L365-372 of the original manuscript, would help the user. Moreover, we made the two end user recommendations more specific: "*For instance, the number of grid cells with a large ratio between gauge and radar accumulations could be counted. If the total count exceeds a threshold, the interval can be labelled as suspicious.*" and "*For example, grid cells with annual precipitation clearly above that of local climatological gauge records could be discarded, as well as grid cells with unlikely high values for 24-h precipitation.*".**

**Minor comments**

1. L43-L44: The method of Park et al. (2019) is rather based on a systematic bias adjustment, instead of merging. Hence, the following sentence can be revised (with bold) as "an operational **gauge-adjusted** OPERA-based radar rainfall product for the European Rainfall-InduCed Hazard Assessment (ERICHA) system. ~~employing data from ~1500 rain gauges.5~~ **6** hours for the European Flood Awareness

System (EFAS)". Here, the number of rain gauges can be misleading because the daily bias map has been obtained from valid radar-rain gauge pairs, which may vary from different gauge sources, quality of both radar and gauge estimates, and days of rainfall over available gauge points. In any case, the algorithm can adapt to the use of more gauges.

**We changed this accordingly. We understand that the number of gauges used in the adjustment will vary and that the method of Park et al. (2019) is well suited for handling more rain gauge data.**

2. L45-L46: "The use of this dataset is restricted to EFAS", This is not correct (e.g., the dataset and the adjustment algorithms have continuously been used in the framework of several research projects since 2017) and do not add any relevant information; so, please remove it.

**We removed this sentence. We wanted to make the distinction that EURADCLIM is publicly available and other OPERA-based datasets are not. This is caused by current data policy and not because of restrictions from the side of developers of these datasets.**

3. L50-L51: Please remove "and data from far more rain gauges are available (~7700)". Perhaps the authors can mention that their dataset is built with different gauge sources from Park et al. (2019), but it is not necessary to point out the number of gauges unless adding an interesting research argument. The more gauges available, the better both methods would perform. In fact, the authors describe the gauge network used in the presented methods (in L104-L120), addressing the number of gauges deployed (~7700), which fits better here and shows the distribution clearly as well.

**The reason to mention this is to show that for EURADCLIM far more rain gauges are available because much more time is available to wait for additional (climatological) rain gauge records. Hence, the quality of EURADCLIM is expected to be higher compared to (near) real-time radar datasets involving rain gauges. This is by no means a methodological disadvantage of Park et al. (2019) and we agree that both methods would perform better if more gauges would be available. We rephrased this (in italic font): "Some differences with the study by Park et al. (2019), *who developed a (near) real-time dataset*, are that additional algorithms to remove non-meteorological echoes are applied, and data from far more rain gauges are available *after waiting 1.5 years* (*at most ~7700 instead of at most a few thousand*)."**

4. L56-L57: "Since it could also be applied in (near) real-time, its evaluation is also relevant for the existing gridded OPERA products". Can this be better clarified?
**We will modify this accordingly (additions in italic font): *The Gabella clutter filter does not depend on historical or auxiliary data and* could be applied in (near) real-time*. Hence, evaluation of its performance to remove non-meteorological echoes* is also relevant for the existing gridded OPERA products*.**

5. L76: "It is based on the raw single site radar data, which have undergone Doppler clutter filtering." Can this part be better explained?
**We will modify this (changes in italic font): "It is based on the *3D* single site radar data *from NMHS*, which have undergone Doppler clutter filtering. *The latter helps to detect and correct for clutter in case the radial Doppler velocity is (near) zero. Hence, the influence of (nearly) stationary targets on radar reflectivity factors is diminished, while preserving the non-stationary precipitation targets.*"**

6. L90-L95: Are the values of "nodata" and "undetect" directly from the OPERA 15-minute rain rate product? Or when calculating the 1-h precipitation accumulation, those values are assigned to the EURADCLIM data cell if not satisfying the full availability (similarly done as the OPERA data)? Here, can this full availability be better explained?
**In case a grid cell has a "nodata" value in at least one of the original**

OPERA 15-min rain rates, the corresponding 1-h accumulation becomes "nodata", which is consistent with OPERA processing. "undetect" values are only obtained from the OPERA 15-min rain rate product and are set to 0. We only assign "nodata" values ourselves when the availability is lower than 83.3% for a 1-h, 24-h or annual precipitation accumulation. We will clarify this in Section 2.1 (additions in italic font): "So*, 1-h accumulations are only derived if all 15-min OPERA rain rates are not equal to the "nodata" value.* And "undetect" values *from the OPERA rain rates*, 'used to denote areas below the measurement detection threshold (radiated but nothing detected)' (Michelson et al., 2019), are set to 0 mm."

7. Fig. 2b, and L107: What does "Combined radar-gauge data availability (in %)" mean?

   **We will clarify this in Section 2.2 by adding (italic font): "*This availability is the percentage of daily intervals in the 8-year period at each gauge location where the gauge and corresponding radar accumulations are both available.*"**

8. L103: Is it necessary to state "mid-June 2022" here? It is actually mentioned better in L115.

   **We agree, and removed this. It is already mentioned elsewhere: "At the time of data production (mid June 2022)".**

9. L121-L130: The authors comment that there are some uncertainties on the gauge aggregation time for the given ECA&D 24-h accumulation dataset and explain that "The end times of the observations display a large variability" in L127. Then, is such variability considered in the disaggregation of the 1-hour dataset applied in L199-L206?

   **The procedure is as follows: first the exact daily measurement interval is determined based on contacting NMHS and by comparing all daily accumulations at gauge locations to those from radar over the 8-year period. This implies that we are confident about the daily measurement interval. Then this daily interval is employed for the disaggregation. So, the variability in end times is taken into account in the disaggregation. We will add to the methodology section (italic font): "To achieve this, the daily gauge accumulations are disaggregated to 1-h accumulations employing the 1-h and 24-h radar accumulations from the previous processing step. *The end times of the observations, as determined in Section 2.2, are taken into account in the disaggregation.*"**

10. Fig.3: For disaggregation, are the 24-h radar precipitation composites used?

    **Yes. This is already stated inside Figure 3: "Convert daily ECA&D gauge data to 1-h data using 1-h and 24-h radar precipitation composites (disaggregation)." and in Section 3.4: "To achieve this, the daily gauge accumulations are disaggregated to 1-h accumulations employing the 1-h and 24-h radar accumulations from the previous processing step."**

11. L205: "at most 4 hours per 24-h interval", does this threshold of 4 hours come from the observed results? I am a bit puzzled, there were days with missing hours of more than 4 hours (as in the case of Fig. 8.a), and if this is the case, how would the correction(merging) be applied?

    **We will add this explanation to Section 3.4: "The threshold of 4 hours comes from the minimum required availability of 83.3% to aggregate 1-h to 24-h radar accumulations. If this requirement is not met for a given radar grid cell and 24-h interval, the "nodata" value is assigned to the radar grid cell. Hence, if more than 4 hours of radar data at the location of a rain gauge are missing per 24 h, the disaggregation to 1-h gauge values cannot be performed. The merging of 1-h disaggregated rain gauge data and radar data is simply performed with all available radar-gauge pairs. If no radar-gauge pairs would be available at all, due to either missing radar (i.e., only "nodata" values at gauge locations) or rain gauge data, the merging is not performed, and EURADCLIM will be a copy of the unadjusted radar dataset.". Note that Fig. 8a contains a rainfall rate that has been used to construct EURADCLIM, and is hence not an example of a day with missing radar data.**

12. Fig. 5 a-d: Maps of mean hourly precipitation over 2013-2020. The mean values are so low, which is not so evident to see the effects over some areas. Is it meaningful to present in terms of mean 1-h precipitation to highlight the effect of clutter filters?

    **The numbers are low indeed, but spatial differences in mean rainfall can be noticed. We think it is important to keep this figure, because it clearly highlights areas influenced by clutter. In our experience, the combined**

use of (average) accumulations and relative exceedance frequencies provides valuable information on the occurrence of clutter and of the effectiveness of algorithms to remove clutter (e.g., Overeem et al., 2020).

13. L271-L273: Although the authors state that there has been a strong reduction in the coastal area of Norway (5a-5d), it is not clear to see. Can it be explained better?

**This is indeed not evident for the entire coastal area of Norway, but only for the southwestern coastal area of Norway. We will clarify this in Section 4.1 (italic font): "And the area in Europe known for the highest annual precipitation, the *southwestern* coastal area of Norway, also shows a strong reduction, which may point to unwarranted classification of non-meteorological echoes. *This can be seen more clearly in Fig. c in Appendix A1.*"**

14. Table 1: Why the values of mean daily precipitation corresponding to 4 different filters are the same?

**Because the mean daily precipitation is based on the reference dataset, i.e., the ECA&D gauge dataset. The use of the gauge dataset for the merging is mentioned in the table caption.**

15. L289: I guess "rho" here is the correlation coefficient. (Please add the symbol in the table or explicitly state "correlation" next to the symbol in L289).

**We will add the symbol to Table 1 and add a description to the caption of Table 1 of the used abbreviations. Note that "rho" is already defined in Section 3.5.**

16. L297-L300: "This confirms the effectiveness of the algorithms to further remove non-meteorological echoes". Can this be better clarified? For no-rain only (0 mm), the effect of the static mask seems to be none based on daily accumulations on average.

**We will add to Section 4.1: "It remains 0.2 mm*, though,* when also the static clutter filter is applied. *This is likely a result of the static clutter mask being only applied to a small part of the OPERA domain (see, e.g., Figure 4h), which contains a small minority of the rain gauges. As a consequence, positive effects are small when comparing to rain gauges from the entire OPERA domain. Moreover, in contrast to the other clutter filtering algorithms, values are not removed but interpolated from surrounding values. Hence, no clear improvement at the location of the gauges is found for the static clutter mask.*"**

17. L305: Regarding the results of an overestimation of 11%, please indicate explicitly EURADCLIM in the text.

**We will add this.**

18. L311: How do the authors separate between summer and winter cases?

**Summer contains data from the months June, July & August and winter contains data from the months December, January, February (as already indicated in the caption of Figure 9). We will clarify this by adding to Section 4.2 (italic font): "The quality of daily precipitation accumulations is higher in summer *(June, July, August; Fig. 6d)* than in winter (*December, January, February*; *Fig. 6c*)."**

19. Fig.6b-6d: Why there are some systematic "0" or near 0 values of EURADCLIM for all values of gauges?

**It is indeed apparent that these are present for this relatively long, daily accumulation period. This could be caused by representativeness errors, e.g., due to timing differences and differences in measurement volumes, especially when local precipitation gradients are large. In case of fast-moving showers, radar precipitation estimates can become less representative for precipitation at the Earth's surface. Given the relatively long accumulation period, these effects will probably play a limited role. It is more likely that beam blockage and overshooting are an important cause. We selected all data points from Figure 6b for which the daily radar**

**accumulations are smaller than 1 mm and the corresponding gauge accumulations are larger than 20 mm. These 5992 data points belong to 1174 gauge locations, which are visualized below.**

[Figure]

**Many locations are in areas either far away from radars (compare with Figure 1c) and/or in areas which may be affected by beam blockage due to mountains (Austria, Italy and Norway). Since 54% of all data points belong to Italy, this is the region where most of the deviations occur. Note that no Italian radar data are available for OPERA. We will add to Section 4.2 (italic font): "*Apparent are the (near) zero precipitation values for a wide range of rain gauge values. 54% of all data points for which the daily gauge accumulation is above 20 mm and the corresponding EURADCLIM radar accumulation is below 1 (Fig. 6b), are located in Italy, and are likely related to beam blockage or overshooting.*"**

20. L334-335: "After all, the gauges that…in the final EURADCLIM dataset". Can this be better clarified? Does it mean that those gauges are used in the evaluation of the EURADCLIM dataset? Or if these are a part of EURADCLIM, how does this justify (link to) the previous statement?
   **For the leave-one-out statistics (LOOS) results, these gauges have indeed been left out of the analysis and are purely used in the evaluation of the EURADCLIM dataset. EURADCLIM uses all rain gauges, but to provide an independent verification, the gauge-adjustment has been rerun for each gauge location while using all available radar-gauge pairs except the one at the considered gauge location. We will clarify this in Section 4.2 by replacing "In reality, results are expected to be better than found for the independent verification, because the distance to the nearest gauge will be much shorter. After all, the gauges that are left out for LOOS have been used in the final EURADCLIM dataset." by (italic font): "In reality, results are expected to be better *for EURADCLIM* than found for *this* independent verification, because the distance to the nearest gauge will be much**

shorter. *For the LOOS results, the gauge-adjustment has been rerun for each gauge location while using all available radar-gauge pairs except the one at the considered gauge location. Whereas for EURADCLIM, all available radar-gauge pairs are used.*".

21. L337: "For other regions, radar beam-blockage could play a role." – Can this be explained better related to Fig.5 and Fig. 7 (e.g., Is it possible to identify those areas which indeed indicate the presented filter that does not seem to work based on the 8-year dataset? If so, in the comparison with gauges, those areas could have been excluded as well).

**See our reply to comment #23, where we discuss beam blockage related to Figure 9 and where more information on beam blockage locations will be added to the revised manuscript. The underestimates in Figures 5 and 7 in Italy and Austria are consistent with these findings, but cannot be found for northwestern Spain. This shows some limitations of rain gauges in detecting beam-blockage-affected areas, due to the limited coverage of rain gauges. Using gauges to exclude areas is not straightforward, since you need to decide which area they are representative for. Moreover, this would result in gaps in the dataset.**

22. Fig.9: Why not use the same colour scale as those presented in Fig. 5? Can this be done in terms of daily accumulation?

**We used a different color scale for EURADCLIM because the precipitation amounts are, on average, much higher than for the OPERA dataset. Employing the same color scale in Figure 9 as in Figure 5, would result in large areas belonging to the highest class (black), and hence would not match the range in precipitation values of EURADCLIM. Instead, we will adapt the color scale in Figure 5 to that of Figure 9. This will reveal somewhat different features, which will require some modifications in the description of Figure 5 in Section 4.1 (see track changes in the revised manuscript).**

**Plotting daily mean accumulation would result in very similar patterns (only some differences are possible due to differences in availability of 1-h and daily data). Adding the (relative) frequency of exceedance for 24-h precipitation may be relevant but would make our manuscript rather lengthy. Moreover, the 1-h accumulations are likely most important for users. We therefore only provide a figure of 1-h accumulations.**

23. L349-L350: Could it be better explained where those signatures are shown (in Fig. 9a or in Fig 9)? As mentioned in #22, can the results be better explained with respect to Fig. 5?

**We will add to Section 4.2 (italic font): "There are still some signatures of beam blockage** *(e.g., Austria, Italy, northwestern Spain in Fig. 9a),* **probably caused by obstacles near radar sites, and of non-meteorological echoes, such as interferences (***e.g., Bosnia and Herzegovina, eastern Spain in Fig 9a,f,g***)." In addition, we will add to Section 4.3: (italic font): "***When comparing the mean daily precipitation and relative frequencies of exceeding 0.1 mm and 5 mm to those from the unadjusted radar data in Fig. 5, it is clear that these strongly increase after the gauge adjustment. This can also lead to non-meteorological echoes becoming more pronounced (e.g., interferences in eastern Spain in Fig. 9f compared to Fig. 5h).***"**

24. L359: "It is difficult to tell to what extent non-meteorological echoes play a role". It is possible that invalid(strange) radar data (similar to the presented case in Spain) are included in the 8-year statistics over Croatia, Kosovo, Slovakia and Corsica. Have the authors checked those areas in particular?

**We agree that also failures in radar processing may play a role, although the specific cases presented in Figure 8 are not apparent in Figure 9. We**

will add to Section 4.3 (additions in italic font): "It is difficult to tell to what extent non-meteorological echoes *or failures in radar processing* play a role here". Figure 9g,h is meant to illustrate the occurrence of extreme values in EURADCLIM. Because the figures are based on processing a large number of events over a wide spatial domain, studying the causes of outliers becomes a topic on its own and hence falls outside the scope of this study.

25. L407-L409: This conclusion needs better supporting materials or references; Because the presented method is based on neither sub-daily gauge data nor a real-time feed environment, it is not clear yet what aspects can be improved. For production in real-time, better improvements in L365- L375 (Major comment 1) are expected.

**These lines should be seen as a recommendation to make it (technically) feasible that NMHS provide sub-daily rain gauge data in (near) real-time for merging with radar data. Then a (near) real-time adjustment could be performed, instead of employing a gauge adjustment based on daily data from the last seven rainy days (Park et al, 2019). We think the quality of radar precipitation estimates will show larger improvements when radar data are merged with rain gauge data from the same time interval than when additional, mostly statistical, algorithms would be applied to further remove non-meteorological echoes (L365-L375 in the original manuscript). Most of the suggested algorithms in L365-L375 are a bit coarse and could be seen as pragmatic ways to deal with the current OPERA radar data. One of the main recommendations in the conclusions section is to apply additional algorithms to volumetric radar data, which can for sure result in large improvements in terms of clutter removal and quantitative precipitation estimation. Note that the conclusions section now provides a list of recommendations for end users, based on L365-L372 in the original manuscript, and a list of recommendations for NMHS, OPERA and EURADCLIM to improve OPERA-based precipitation products. For the latter, the contents of L372-L375 from the original manuscript was added.**

26. Code availability: It is good that the authors provide some routines, but this code will not be sufficient to "reproduce" the results and the end-users required a good knowledge of the OPERA data structure. So, rephrase or remove "This helps end users to reproduce results from this study and to further explore and analyze the EURADCLIM dataset."

**The code can be used to reproduce the figures, i.e., to visualize precipitation maps. Indeed, code to reproduce EURADCLIM is not provided. It can be used to explore and analyze the EURADCLIM dataset, though, by performing climatological analyses, aggregating data and visualization. We rephrased this: "This can help end users to visualize precipitation maps and to further explore and analyze the EURADCLIM dataset".**

27. Data availability: Hourly dataset is useful, however, if the 24-h precipitation datasets are simply generated by summing up hourly data in running windows, is there any specific reason to produce such big-size outputs with a separate reference?

**This prevents many users from doing the same aggregation, and is hence provided as an additional service. See also our response at the beginning of this rebuttal.**

**References**

**Overeem, A., Uijlenhoet, R., and Leijnse, H.: Full-year evaluation of non-meteorological echo removal with dual-polarization fuzzy logic for two C-band radars in a temperate climate, J. Atmos. Oceanic Technol., 37, 1643–1660, https://doi.org/10.1175/JTECH-D-19-0149.1, 2020.**

---

## Author Response (AR2)

**Dear Topical Editor, dear Alexander Gruber,**

**Thank you for conditionally accepting our manuscript. Please find below a brief response to the minor comments addressed by Referee #2 in order for you to know what we did with the comments.**

**Yours sincerely,**
**Aart Overeem**
**Royal Netherlands Meteorological  Institute**

Topical Editor decision

Dear authors,

thank you for the revision of your manuscript, which addressed all concerns raised by the referees properly. Only Referee #2 has is asking for two small clarifications. Since these are very minor, I am willing to accept your manuscript and leave addressing these comments up to your discretion. I am sure your paper will make a valuable contribution to the community.

Best regards,
Alexander Gruber

Referee #2

I would recommend its publication after considering a couple of minor comments below.

Minor comments:
1. Introduction L53-67: This part intends to address the motivation/objectives of what has been done in EURACLIM and what for. However, it still lacks clarity in writing (e.g., because of addressing the possibility of a "real-time" application of Gabella clutter filter in the introduction, I expected to see the results produced in real-time for the filtering. Was the presented EURACLIM dataset generated in such a way applying the real-time filter? If not, this creates confusion. Indeed, L427-L432 does mention the possibility of applying it in (near) real-time in the context of future applications as a conclusion. Also, some detailed setup/statements regarding cloud type, E-OBS, and computation of clutter mask seem to suit better as the "summary" in the conclusion section. So, a revision/rephrasing can help readers.
**We clarified the part in the Introduction referring to a possible real-time application of the Gabella clutter filter. We think it is better to keep the setup/statements. In our opinion, these are not too detailed, but help the reader right from the beginning to understand the basic processing of EURADCLIM.**

2. Table 1: As the authors answered (#14-minor comments), if the comparisons are done based on the reference dataset (the same pair of radar and gauge for four different filters), the number of pairs (indicated "n", right? Please add its description in the table as well) for each threshold should be the same for different filters (as seen in the mean). Why is the n different?
**We clarified Table 1 accordingly. The differences in the value of n were almost entirely related to slightly different verification periods. Now we select exactly the same period as for the**

**EURADCLIM datasets. There are still some differences in the number of radar-gauge pairs, but now they have become very minor.**